# What Matters in RL-Based Methods for Object-Goal Navigation? An Empirical Study and A Unified Framework

## Abstract

Object-Goal Navigation (ObjectNav) is a critical component toward deploying mobile robots in everyday, uncontrolled environments such as homes, schools, and workplaces. In this context, a robot must locate target objects in previously unseen environments using only its onboard perception. Success requires the integration of semantic understanding, spatial reasoning, and long-horizon planning, which is a combination that remains extremely challenging. While reinforcement learning (RL) has become the dominant paradigm, progress has spanned a wide range of design choices, yet the field still lacks a unifying analysis to determine which components truly drive performance. In this work, we conduct a large-scale empirical study of modular RL-based ObjectNav systems, decomposing them into three key components: perception, policy, and test-time enhancement. Through extensive controlled experiments, we isolate the contribution of each and uncover clear trends: perception quality and test-time strategies are decisive drivers of performance, whereas policy improvements with current methods yield only marginal gains. Building on these insights, we propose practical design guidelines and demonstrate an enhanced modular system that surpasses State-of-the-Art (SotA) methods by 6.6% on SPL and by a 2.7% success rate. We also introduce a human baseline under identical conditions, where experts achieve an average 98% success, underscoring the gap between RL agents and human-level navigation. Our study not only sets the SotA performance but also provides principled guidance for future ObjectNav development and evaluation.

## 1 Introduction

Recent advances in computer vision and deep learning have inspired growing interest in interdisciplinary applications that bridge perception, reasoning, and control, especially in robotics. Among these, vision-based navigation has emerged as a foundational capability for autonomous mobile agents. A key benchmark in this domain is *Object-Goal Navigation (ObjectNav)*, where a robot must navigate to an instance of a specified object category in an unseen environment, relying solely on its onboard sensors. This task is both practically important and technically challenging: it requires semantic understanding, spatial reasoning, and long-horizon planning. Among many approaches, Reinforcement Learning (RL) has become a dominant paradigm for ObjectNav, offering a structured framework to learn directly through trial-and-error and showing steady progress across various benchmarks. While end-to-end RL policies are common, modular RL approaches have shown greater robustness and improved generalization. By decomposing the system into interpretable and tunable components, such as perception, mapping, policy, and action execution, these methods align with the multi-faceted nature of ObjectNav and often achieve better sim-to-real transfer (Gervet et al., 2023). However, this modular design also increases overall system complexity, and the standalone contribution of each component has not been systematically studied in recent literature. As a result, current bottlenecks remain unclear, and the lack of a unified design guide makes it difficult to fairly evaluate and advance modular RL systems.

In this work, we disentangle the components of modular RL-based ObjectNav and ask a fundamental question: *Which design choices truly matter for RL-based ObjectNav?* We establish a principled decomposition of modular ObjectNav systems into three essential parts—perception, policy,

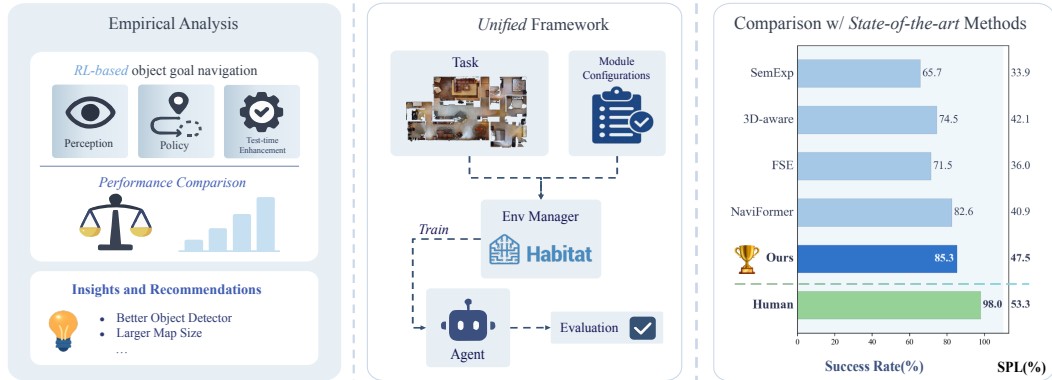

Figure 1: Overview of our work. Our framework encompasses: (1) an empirical study analyzing the impact of different modules, and (2) a unified framework with interchangeable components, enabling users to customize their own object-goal navigation policies.

and test-time enhancement. For each component, we categorize representative design choices from the literature and, through extensive experiments, isolate the individual contribution of each, an overview of our approach and results is demonstrated in Fig. 1.

Our key findings are as follows: (a) The capability of the perception module has a substantial impact on overall navigation performance. (b) Test-time enhancement techniques are often overlooked; however, they play a surprisingly significant role in boosting performance. (c) With well-designed perception and test-time modules, further improvements must come from the policy module; however, we observe that with current learning approaches, such gains are limited or marginal.

Based on these findings, we offer practical recommendations for selecting and combining system components, along with insights into the reasoning behind them. Following these guidelines, we design an enhanced modular system that achieves state-of-the-art performance with 47.5% SPL (success weighted by path length) and 85.3% success rate, surpassing the best prior methods by 6.6% in SPL and 2.7% in success rate. In addition, we introduce a human baseline comparison, where expert participants operate under the same training setup, test environments, and observation modalities as the agent, achieving an average of 98.0% success rate and 53.3% SPL. This contrast reveals a clear gap between current RL-based systems and human-level navigation, emphasizing the need for new algorithms that can enhance both performance and robustness in ObjectNav.

Instead of emphasizing our empirical gains, our primary goal is to highlight broader implications for the vision-based navigation community from the design of future algorithms to the establishment of fair evaluation protocols and the deployment of ObjectNav systems. We will publicly release our code and evaluation tools to the community.

## 2 RELATED WORK

**End-to-end Object-Goal Navigation.** These approaches leverage reinforcement learning or imitation learning to train a policy that directly maps visual observations to low-level actions. Early works by Ye et al. (2021); Ramrakhya et al. (2022; 2023) followed the architectural blueprint of DD-PPO (Wijmans et al., 2020), employing convolutional neural networks (CNNs) coupled with recurrent neural networks (RNNs). Subsequent methods (Yadav et al., 2022; Khandelwal et al., 2022) enhanced this framework by replacing the RGB encoder with more powerful self-supervised vision models, leading to improved performance. More recent efforts (Zeng et al., 2024; Ehsani et al., 2024) have adopted transformer-based policy architectures to better capture long-term spatial dependencies, achieving state-of-the-art results. To further enrich spatial reasoning, several works have proposed integrating structured scene representations Gadre et al. (2022) into the policy network. These methods (Yang et al., 2018; Qiu et al., 2020; Zhou et al., 2021; Guo et al., 2021; Zhang et al., 2021) build scene graphs from RGB inputs, encode them with Graph Neural Networks (GNNs), and feed the features into the policy. One major downside of end-to-end methods is that they require large-scale data and struggle to generalize across the sim-to-real gap (Gervet et al., 2023).

**Modular RL Methods for Object-Goal Navigation.** These approaches integrate learning-based components with classical map-based navigation to achieve improved data efficiency, lower computational overhead, and competitive performance. One representative approach is ANS by Chaplot et al. (2020b), which employs a reinforcement learning-based pipeline for navigation in 3D environments. ANS consists of a mapping module that projects RGB-D observations into a top-down occupancy map, along with a long-term goal policy trained via reinforcement learning to guide exploration. Building on ANS, SemExp (Chaplot et al., 2020a) introduces a semantic mapping module that augments the top-down map with object-level semantic annotations. Subsequent work largely follows this modular reinforcement learning framework with various enhancements. For instance, FSE by Yu et al. (2023a) incorporates frontier detection into the mapping process and selects subgoals based on the centers of frontier regions. 3D-Aware (Zhang et al., 2023) further utilizes 3D structural cues to improve scene understanding and predicts corner-based sub-goals, inspired by heuristic strategies like those in Luo et al. (2022). More recently, NaviFormer (Xie et al., 2025) achieves state-of-the-art performance by replacing traditional map encoders with a transformer-based architecture, enabling richer spatial-semantic representation learning. While existing modular methods primarily focus on designing more effective navigation policies, they often overlook a systematic analysis of how individual components within each module affect overall system performance. In this work, we present a comprehensive study of modular architectures, explicitly evaluating the role and contribution of each component to the navigation pipeline.

## 3 STUDY DESIGN

**Problem Formulation.** In the context of ObjectGoal navigation, the agent is required to navigate to an instance of a specified object category within an unseen environment. At each timestep $t$, the agent receives an RGB-D observation $o_t \in \mathbb{R}^{4 \times H \times W}$, a goal category label $g$, and its current pose $p_t$. Based on these inputs, the agent generates actions to move towards the goal. We use Habitat (Puig et al., 2023) as a unified simulation testbed, which has a built-in low-level controller that outputs one of four discrete actions: `move forward`, `turn left`, `turn right`, and `stop`.

**System Overview.** Figure 1 outlines the structure of our study and analysis. We follow the natural pipeline of robot navigation and decompose a typical RL-based approach into three core modules: perception, policy, and test-time enhancement. The perception module covers the process from raw sensor inputs to building representations of the environment. The policy module takes the output of the perception module and predicts the next action based on its action space. The test-time enhancement module contains inference-time strategies that improve performance during execution by adjusting the agent's behavior without retraining. These strategies operate alongside the learned policy to correct common failure patterns and strengthen the overall perception–action loop.

For each of these three components, we survey a wide range of design choices found in recent literature and conduct extensive experiments under controlled settings to isolate and quantify their individual contributions to the final navigation performance.

**Perception Module Design Choices.** A typical representation built by the perception module is a top-down semantic map, which captures both geometry and semantic information based on historical input images. At each timestep, an object detector extracts semantic labels from RGB images, which are then fused with depth to build

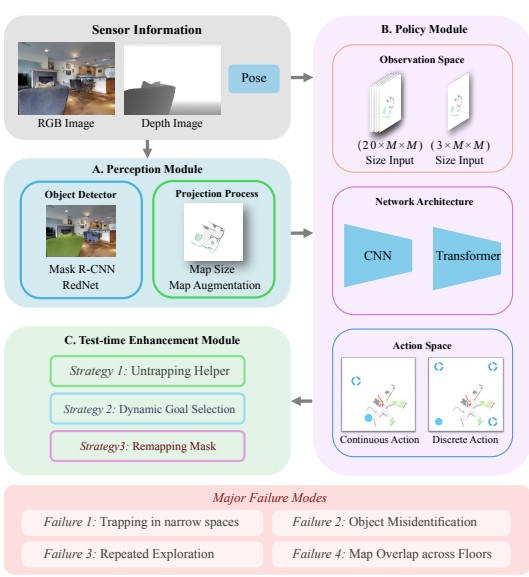

Figure 2: Unified Framework of our experimental setting. *A. Perception:* RGB-D and pose are fused into a top-down semantic map. *B. Policy:* The map and auxiliary inputs (e.g., category, orientation) guide action prediction. *C. Test-Time Enhancement:* Plug-and-play strategies applied at evaluation to boost performance without retraining. The figure also highlights the major failure modes addressed by these strategies.

a structured 3D representation. Aggregating this representation along the height axis yields a 2D semantic map suitable for navigation.

In our study, we focus on three widely adopted design choices for the perception module—*object detector*, *map augmentation*, and *map size*, as these have been repeatedly emphasized in prior ObjectNav systems (Chaplot et al., 2020b;a; Yu et al., 2023a; Zhang et al., 2023; Xie et al., 2025). A detailed description of these components is provided in Appendix A.10.

**Policy Module Design Choices.** With the top-down semantic map in place, the policy module infers a goal location that guides the agent's navigation. Typical RL-based policy for long-term goal prediction and identification consists of four key design choices: the *observation space*, the *action space*, the *network architecture*, and the *reward design*. Each of these components influences how the agent interprets the semantic map and decides on navigation strategies. In our experiments, we adopt the common setup established in prior work (Chaplot et al., 2020b;a; Yu et al., 2023a; Xie et al., 2025; Wu et al., 2022). We refer the readers to Appendix A.11 for the implementation details.

**Policy Learning Algorithm.** We adopt PPO Schulman et al. (2017) as the default learning algorithm for the policy module because it is the dominant choice in prior ObjectNav work and consistently offers stable performance under sparse rewards and long-horizon navigation. This observation is consistent with broader trends in robotics RL, where PPO is the dominant choice across quadrupedal locomotion Lee et al. (2020); Hwangbo et al. (2019), agile aerial navigation Xing et al. (2024b;a), and humanoid control Radosavovic et al. (2024) due to its reliability and ease of tuning. To illustrate how policy-learning choices affect performance, we include a small ablation on PPO's clipping threshold (Table 7). The standard setting ($\epsilon = 0.2$) clearly outperforms an extremely restrictive value ($\epsilon = 0.001$), which severely limits learning progress. This highlights the importance of maintaining a reasonable update range for effective policy optimization.

**Test-time Enhancement Module Design Choices.** Common failure modes found during evaluations are categorized into: (a) trapping in narrow spaces, (b) object misidentification, (c) repeated exploration, and (d) map overlap across floors due to undetected staircases (Figure 2). To mitigate these issues, several strategies have been used in previous work, but were not formally documented. We summarize them into three plug-and-play strategies: (i) an *untrapping helper* to prevent the agent from getting stuck, (ii) *dynamic goal selection* to avoid redundant exploration, and (iii) a *remapping mask* to handle multi-floor map overlap. We refer readers to Appendix A.12 for further details.

## 4 EXPERIMENTS, RESULTS, AND RECOMMENDATIONS

Following our definition of three main modules and different design choices for each one, we organize our experiments by analyzing the contribution of individual design choices to overall navigation performance. For each experiment, we provide a detailed setup and report quantitative results, followed by in-depth interpretation and analysis. Alongside our findings, we offer actionable insights and practical recommendations, which can serve as a foundation for designing effective pipelines in future research.

To evaluate the performance of the proposed methods, we consider two evaluation settings: *Fixed timestep* and *Dynamic timestep*. In the fixed-timestep setting, the agent is allowed to interact with the environment for up to 500 steps. In the dynamic timestep setting, the maximum number of steps is set proportional to the number of steps required by an optimal planner to reach the goal. In both cases, the task is considered successful if the agent is within 1 meter of the target object instance at the end of the episode. Further details are provided in Appendix A.9.

We adopt three standard evaluation metrics proposed by Anderson et al. (2018): Success Rate (SR): the ratio of successful episodes to the total number of episodes. Success weighted by Path Length (SPL): the ratio between the shortest-path distance and the actual path length taken by the agent, averaged over successful episodes. Distance to Success (DTS): the agent's distance to the target object at the end of an episode. In the Dynamic timestep setting, we report the corresponding variants: D-SR, D-SPL, and D-DTS.

### 4.1 EFFECT OF PERCEPTION MODULE

To isolate the influence of the perception module itself, we implement two heuristic-based long-term goal policies that use heuristic rules for navigation instead of a neural network policy conditioned on the top-down semantic map. This allows us to evaluate the Perception Module independently of long-term goal policy effects.

Table 1: Study results of perception module for Corner (C) and Frontier (F) goal policies. MRCNN denotes Mask R-CNN with the default checkpoint, while FT-MRCNN denotes Mask R-CNN with the Gibson-finetuned checkpoint.

| Pol. | Det. | Size | Aug. | SR (%) | SPL (%) | DTS (m) | D-SR (%) | D-SPL (%) | D-DTS (m) |
|---|---|---|---|---|---|---|---|---|---|
| C | MRCNN | 480 | ✓ | 77.7 | 40.9 | 1.047 | 61.6 | 38.6 | 1.641 |
| C | RedNet | 480 | ✓ | 80.9 | 43.9 | 0.851 | 65.5 | 41.6 | 1.488 |
| C | RF-DETR-Seg | 480 | ✓ | 77.6 | 44.2 | 0.991 | 64.1 | 42.4 | 1.512 |
| C | RF-DETR+SAM2 | 480 | ✓ | 77.1 | 44.7 | 1.112 | 63.8 | 42.6 | 1.669 |
| C | YOLO11-XL | 480 | ✓ | 76.0 | 42.6 | 1.147 | 62.8 | 40.8 | 1.668 |
| C | FT-MRCNN | 480 | ✓ | 83.8 | 47.6 | 0.800 | 69.5 | 45.3 | 1.425 |
| C | FT-MRCNN | 240 | ✓ | 78.7 | 45.2 | 1.135 | 64.8 | 43.1 | 1.658 |
| C | FT-MRCNN | 480 | ✗ | 83.3 | 44.2 | 0.771 | 63.5 | 41.2 | 1.727 |
| F | MRCNN | 480 | ✓ | 71.2 | 38.5 | 1.428 | 49.1 | 33.5 | 2.086 |
| F | RedNet | 480 | ✓ | 75.0 | 40.9 | 1.114 | 53.1 | 35.7 | 1.876 |
| F | FT-MRCNN | 480 | ✓ | 79.9 | 45.8 | 0.892 | 57.2 | 39.7 | 1.714 |
| F | RF-DETR-Seg | 480 | ✓ | 72.9 | 40.3 | 1.183 | 53.6 | 36.6 | 1.871 |
| F | RF-DETR+SAM2 | 480 | ✓ | 72.8 | 43.7 | 1.416 | 53.6 | 39.2 | 2.006 |
| F | YOLO11-XL | 480 | ✓ | 69.5 | 39.0 | 1.514 | 49.8 | 34.6 | 2.106 |
| F | FT-MRCNN | 240 | ✓ | 79.6 | 43.6 | 1.033 | 61.0 | 40.1 | 1.742 |
| F | FT-MRCNN | 480 | ✗ | 78.0 | 42.5 | 1.000 | 53.4 | 36.3 | 1.791 |

(i) *Corner Goal Policy*: This method adopts a similar action prediction strategy as Stubborn (Luo et al., 2022). Instead of selecting goals in a deterministic or fixed pattern, we introduce randomness into the action selection process, similar to MOPA (Raychaudhuri et al., 2024). This enables more exploratory behavior in the environment, allowing us to better evaluate the influence of components in the Perception Module under varied navigation conditions.

(ii) *Frontier-Based Policy*: As there is currently no standard implementation of the Frontier-Based Policy (Yamauchi, 1997), we adopt a similar setup to FSE (Yu et al., 2023a), but replace the learned policy with a random action policy. Since FSE already includes strong heuristic mechanisms for frontier selection, randomly selecting a goal from the list of frontiers serves as a reasonable approximation of the classical Frontier Exploration method.

### 4.1.1 OBJECT DETECTOR

**Study description.** The object detector takes an RGB or RGB-D image as input and outputs the semantic segmentation of the image. It has a dominant impact on overall performance, as the semantic map depends on its output and it directly serves as input to the policy module. We summarize all the object detectors used in the prior works: *Mask R-CNN with Default Checkpoint*: This variant uses the R50-FPN checkpoint from the Detectron2 model zoo (Wu et al., 2019) without any domain-specific fine-tuning. *Mask R-CNN with Gibson-Finetuned Checkpoint*: This checkpoint, originally introduced in PONI (Ramakrishnan et al., 2022), is fine-tuned on the Gibson dataset (Xia et al., 2018) to better adapt to the domain characteristics of indoor navigation environments. *RedNet*: A network specifically designed for indoor RGB-D semantic segmentation. It is fine-tuned on 100K randomly sampled views from the MP3D (Chang et al., 2017) dataset. This model was first utilized in Stubborn (Luo et al., 2022). Additionally, we evaluate two recent object detectors, *YOLOv11* (Jocher & Qiu, 2024) and *RF-DETR* (Robinson et al., 2025), and further explore combining them with the foundation model *SAM2* (Ravi et al., 2024) to obtain segmentation masks through an alternative approach.

**Interpretation.** From Table 1, we observe that both the Corner Goal Policy and Frontier-Based Policy experiments exhibit similar trends. More advanced perception model generally yields better performance. And a fine-tuned object detector consistently leads to an obvious performance boost across all evaluation metrics. Specifically, for the Corner Goal Policy, using a fine-tuned Mask R-CNN model results in a 6.8% increase in Success Rate, a 6.7% improvement in Success weighted by Path Length, and a 0.247m reduction in Distance to Success. Similar improvement occurs when using our Dynamic metrics. Additionally, although RedNet is not trained on Gibson, fine-tuning on indoor scenes also helps reduce distribution errors. More detailed performance analysis can be found in Appendix A.14.

**Recommendation.** Unsurprisingly, improving the object detector yields significant gains in overall navigation performance. For real-world use, it is crucial to choose a detector trained on data that closely reflects the domain of your test scenes. When such alignment is not available out-of-the-box, collecting representative data and fine-tuning an existing model can significantly improve both detection accuracy and overall navigation robustness.

### 4.1.2 MAP SIZE

**Study description.** Map size defines the agent's field of view. Larger maps reveal more of the environment, but increase computational cost. To investigate its effect, we experiment with two settings: $240 \times 240$ and $480 \times 480$. We focus on the local ego-centric map, as the global map is only used to update the local map and is not directly used for navigation. Therefore, when we refer to the top-down semantic map in this context, we mean the local top-down semantic map.

**Interpretation.** From Table 1, we observe that for the Corner Goal Policy, a larger top-down semantic map generally yields better performance than a smaller one. However, this trend does not hold for the Frontier-Based Policy. In this setting, the performance differences between the two map sizes are minimal. In fact, for some Dynamic metrics such as D-SR and D-SPL, the smaller map size slightly outperforms the larger one. We find that map size influences policies differently: larger maps encourage the Corner Goal Policy to explore wider areas and thus improve performance, while the Frontier-Based Policy is less sensitive to map size and can even degrade with higher-resolution maps due to frontier extraction errors. Details are provided in Appendix A.14.

**Recommendation.** Map size should be chosen based on the policy type. For goal-based policies like the Corner Goal Policy, larger maps encourage broader exploration and generally improve performance. For frontier-based policies, map size has minimal impact and may even degrade performance due to resolution artifacts. Smaller maps also reduce memory and computation, making them preferable in resource-constrained settings.

### 4.1.3 MAP AUGMENTATION

**Study description.** Map augmentation typically refers to the techniques applied during the projection from 3D voxels to 2D maps (Yu et al., 2023a;b; Xie et al., 2025). It often involves fine-grained enhancements that aim to make the top-down map more accurately reflect the real-world scene. To study its impact, we conduct two sets of experiments with and without doing augmentation.

**Interpretation.** As shown in Table 1, Under the standard evaluation setting, both the Corner Goal Policy and the Frontier-based Policy achieve slightly better or comparable performance when map augmentation is applied. However, under our proposed evaluation framework, map augmentation leads to a significant improvement in performance for both policies. We find that map augmentation mainly improves the quality of the constructed maps, enabling richer and more accurate representations within the same number of steps. This leads to clear gains under stricter evaluation, while the benefit diminishes under standard evaluation where agents have more time to explore (see Appendix A.14 for details).

**Recommendation.** Map augmentation generally leads to better performance compared to not using it. However, it often depends on extensive engineering experience and is typically tailored for fixed robotic settings. For more practical or time-sensitive applications, where quick adaptation and reliable perception are critical, applying map augmentation is strongly recommended. Conversely, in generalized scenarios or applications that do not require rapid task completion, operating without map augmentation can also be a reasonable and efficient choice.

## 4.2 EFFECT OF POLICY MODULE

The policy module plays a central role in predicting goals for the agent to reach, and its decision-making process directly influences the agent's ability to explore the environment effectively. To better isolate the contribution of different components within the policy module, we fix the parameters of the perception module. Additionally, when analyzing the effect of a single component, we keep all other factors constant.

Table 2: Study results of key architectural and training choices, including observation/action spaces, network architectures, and reward design.

| Observation Space | Action Space | Network Arch. | Reward Design | SR (%) | SPL (%) | DTS (m) | D-SR (%) | D-SPL (%) | D-DTS (m) |
|---|---|---|---|---|---|---|---|---|---|
| Standard | Discrete | Transformer | Type 2 | 83.4 | 43.8 | 0.717 | 66.3 | 41.0 | 1.501 |
| Compressed | Discrete | Transformer | Type 2 | 83.8 | 43.3 | 0.685 | 65.8 | 40.3 | 1.572 |
| Compressed | Continuous | Transformer | Type 2 | 84.4 | 48.2 | 0.741 | 69.6 | 45.8 | 1.415 |
| Compressed | Discrete | CNN | Type 2 | 83.7 | 43.3 | 0.706 | 65.8 | 40.5 | 1.548 |
| Compressed | Discrete | Transformer | Type 1 | 85.2 | 45.8 | 0.664 | 67.8 | 43.0 | 1.346 |

### 4.2.1 OBSERVATION SPACE

**Study description.** To investigate the impact of different observation space choices on performance, we consider two options: *Top-down semantic map (standard)*: Following most prior work, this setup uses multiple channels to store different types of information, including the exploration map, obstacle map, frontier map, and semantic maps. *Top-down semantic map (compressed)*: In this setting, we propose to use a compressed version of the top-down semantic map. Specifically, we compress the original 20 channel semantic map into a single 3-channel RGB image.

**Interpretation.** Table 2 shows that the performance differences between the standard and compressed configurations are minimal: less than 0.5% in SR and D-SR, less than 0.7% in SPL and D-SPL, and less than 0.1m in DTS and D-DTS. These negligible differences suggest that the compressed top-down semantic map configuration retains sufficient spatial information for effective policy learning, while reducing the observation size by more than a factor of six.

**Recommendation.** A compressed observation is clearly a more efficient and practical choice.

### 4.2.2 ACTION SPACE

**Study description.** The action space determines how the policy generates goals on the map. We investigate the impact of different action space designs by considering the following two options: *Continuous Action Space*: The policy predicts a goal location directly on the map. The goal can be any coordinate within the map boundaries. *Discrete Action Space*: A set of possible goal locations—typically the four corners of the map—is predefined as candidate actions. The policy then predicts the index of one of these predefined goal locations.

**Interpretation.** The 2nd and 3rd row of Table 2 shows that continuous action space outperforms the discrete one. Compared with the continuous action space, the discrete action space is more akin to predicting a direction for the agent rather than an exact goal. Given the scale of the map, the agent's movement range during local policy navigation is limited and often insufficient to directly reach the final goal. As a result, the selected goal in the discrete setting effectively serves as a directional signal, guiding the agent toward further exploration. Therefore, the discrete action space is coarser compared to the continuous action space, as it provides less precise control over goal selection.

**Recommendation.** Without considering additional strategies to further enhance the discrete action space, the continuous action space is a more favorable choice.

### 4.2.3 NETWORK ARCHITECTURE

**Study description.** Previous works commonly used CNNs to process the observations, more recent studies have shifted to using Transformers. Specifically, in our study, we choose: *CNN*: For the CNN-based model, we adopt ResNet-18 (He et al., 2015) as the policy backbone. *Transformer*: For our transformer model, we adopt the ViT-Base architecture (Wu et al., 2020) as the backbone for map feature extraction.

**Interpretation.** The results in Table 2 indicate no significant difference in performance between the two network architectures. These findings indicate that, when keeping other modules unchanged, varying the network architecture for map feature extraction does not substantially affect the final outcomes. This also indicates that the overall performance is sensitive to the policy network as other modules.

Table 3: Study results of test-time enhancement strategies, including untrapping helper, dynamic goal selection and remapping mask.

| Policy | Untrapping Helper | Dynamic Goal Selection | Remapping Mask | SR (%) | SPL (%) | DTS (m) | D-SR (%) | D-SPL (%) | D-DTS (m) |
|---|---|---|---|---|---|---|---|---|---|
| RL (Continuous) | ✗ | ✓ | ✓ | 81.3 | 47.3 | 0.741 | 69.1 | 45.3 | 1.509 |
| RL (Continuous) | ✓ | ✓ | ✗ | 82.6 | 48.0 | 0.817 | 70.1 | 45.9 | 1.384 |
| RL (Continuous) | ✓ | ✓ | ✓ | 84.4 | 48.2 | 0.741 | 69.6 | 45.8 | 1.415 |
| RL (Discrete) | ✓ | ✗ | ✓ | 83.8 | 43.3 | 0.685 | 65.8 | 40.3 | 1.572 |
| RL (Discrete) | ✓ | ✓ | ✓ | 85.3 | 47.5 | 0.632 | 69.8 | 44.9 | 1.397 |

**Recommendation.** Since there is no significant difference in performance between CNN and Transformer backbones, we suggest choosing the architecture based on practical constraints. For real-world or resource-constrained settings, a lightweight CNN (e.g., ResNet-18) is a more efficient and effective choice without compromising performance.

### 4.2.4 REWARD DESIGN

**Study description.** Reward design is always a critical challenge for reinforcement learning algorithms. In this study, We consider two types of reward designs: $r_1 = r_{\text{exploration}} + r_{\text{distance to target}}$, and $r_2 = r_{\text{exploration}} + r_{\text{success}} + r_{\text{step penalty}}$.

**Interpretation.** Table 2 shows that the policy trained with $r_1$ outperforms that trained with $r_2$, yielding a 0.8% improvement in Success Rate and a 2.5% improvement in SPL. With $r_1$, the agent is rewarded for moving closer to the target object, with semantic information implicitly embedded in the reward signal. In contrast, $r_2$ is designed solely for exploration, where only task success is rewarded and additional steps are penalized; the agent receives no reward for approaching the target.

**Recommendation.** Based on the results, when designing the reward for policy training, adding a reward for moving closer to the target object encourages the agent to make more efficient decisions, not only for exploration but also for approaching the target.

### 4.3 EFFECT OF TEST-TIME ENHANCEMENT MODULE

To further improve the robustness and accuracy of the methods, several test-time strategies are commonly employed. These strategies are often overlooked in discussions, as they are considered more of an engineering detail than a scientific contribution. In this section, we aim to clearly investigate the impact of these strategies on overall performance.

### 4.3.1 UNTRAPPING HELPER.

**Study description.** The untrapping helper uses predefined rules to help the agent escape from stuck situations. To assess its impact, we compare our continuous-action RL policy with and without the helper enabled.

**Interpretation.** As shown in Table 3, the use of the untrapping helper clearly improves performance in both evaluation settings. The improvement is more obvious in the standard evaluation. Furthermore, we observe that the untrapping helper has a greater impact on long-distance navigation. When the target goal is farther from the agent, the likelihood of the agent getting trapped increases due to the extended navigation path. This explains why the performance improvement is more pronounced in the standard evaluation setting.
**Recommendation.** The untrapping helper can be applied to various policies to enhance long-term navigation performance.

### 4.3.2 DYNAMIC GOAL SELECTION

**Study description.** Dynamic goal selection is a strategy originally designed for heuristic discrete policies to prevent the agent from engaging in redundant exploration. We further demonstrate that it can also be incorporated into RL-based discrete action space policies to guide the local policy more efficiently. To assess its impact, we compare the performance of our trained discrete policy with and without this mechanism.

**Interpretation.** As shown in Table 3, the improvement provided by dynamic goal selection is significant. In both evaluation settings, we observe substantial performance gains. This is primarily because dynamic goal selection effectively reduces repeated exploration, thereby enhancing navigation efficiency.

**Recommendation.** In the discrete action space, incorporating dynamic goal selection proves to be a beneficial enhancement, significantly improving both navigation performance and efficiency. However, this dynamic goal selection technique relies on prior experience gained through multiple trials. As a result, applying it to continuous or frontier-based action spaces requires more effort, particularly in defining and tuning the relevant parameters to achieve comparable effectiveness.

### 4.3.3 REMAPPING MASK

**Study description.** The remapping mask assists the agent in regenerating the map when overlapping issues occur during navigation. To evaluate the effectiveness of this mechanism, we perform an ablation study using our trained continuous action space policy.

**Interpretation.** Similar to the effect of the untrapping helper, the remapping mask has a more significant impact in the standard evaluation setting, as shown in Table 3, compared to our more stringent setting. This is because long distance navigation increases the likelihood of the agent encountering map overlap issues, such as revisiting previously seen areas from different floors (e.g., via stairs). In contrast, during short distance navigation tasks, if the agent can reach the target object within just a few steps, it is unlikely to traverse into stair regions that could cause overlaps in the top-down semantic map. As a result, the benefit of the remapping mask in these short scenarios is minimal.

**Recommendation.** The remapping mask can be considered a general-purpose enhancement that can be integrated as a plug-in to improve the performance of various navigation policies.

### 4.4 COMPARISONS WITH THE STATE-OF-THE-ART AND HUMAN EXPERTS

After clearly understanding the impact of different components from various modules on overall performance, we selected four policies, each configured with the best-performing components, to compare against prior work. Standard evaluation metrics were used for this comparison.

To enable comparison with state-of-the-art methods, we design four customized policies: (i) Corner Goal Policy, (ii) Frontier-Based Policy, (iii) RL policy with a discrete action space, and (iv) RL policy with a continuous action space. For all policies, we adopt the best-performing configurations across all modules. From Table 4, we observe that after thoroughly analyzing the impact of different components, our configured policy outperforms previous state-of-the-art algorithms on the Gibson benchmark. Since modular methods consist of multiple components that jointly contribute to overall performance, it is essential to understand the impact of each component. Without a clear analysis of these components, focusing solely on designing com-

| Method | SR(%) | SPL(%) | DTS(m) |
|---|---|---|---|
| SemExp (Chaplot et al., 2020a) | 65.7 | 33.9 | 1.47 |
| SemExp* (Ramakrishnan et al., 2022) | 71.7 | 39.6 | 1.39 |
| PONI (Ramakrishnan et al., 2022) | 73.6 | 41.0 | 1.25 |
| 3D-aware (Zhang et al., 2023) | 74.5 | 42.1 | 1.16 |
| FSE (Yu et al., 2023a) | 71.5 | 36.0 | 1.35 |
| SGM (Zhang et al., 2024) | 78.0 | 44.0 | 1.11 |
| L3MVN (Yu et al., 2023b) | 76.8 | 38.8 | 1.01 |
| NaviFormer (Xie et al., 2025) | 82.6 | 40.9 | 0.76 |
| T-Diff (Yu et al., 2024) | 79.6 | 44.9 | 1.00 |
| HOZ++ (Zhang et al., 2025) | 78.2 | 44.9 | 1.13 |
| Corner Goal Policy (Ours) | 83.8 | 47.6 | 0.80 |
| Frontier-based Policy (Ours) | 79.9 | 45.8 | 0.89 |
| Discrete Action Policy (Ours) | 85.3 | 47.5 | 0.63 |
| Continuous Action Policy (Ours) | 84.4 | 48.2 | 0.74 |
| Human Experts | 98.0 | 53.3 | 0.26 |

Table 4: Comparisons with Prior Methods on Gibson. The metrics of the two best-performing methods are highlighted in green and orange. Our method, combined with the findings, demonstrates better performance than all state-of-the-art approaches.

plex policy networks is unlikely to achieve optimal performance. Moreover, complex policy designs often introduce additional challenges, such as increased computational cost and reduced inference speed. Hence, a clear understanding of modular components is essential and can provide valuable insights for future research.

To better contextualize system performance, we introduce a human expert baseline under the same training setup, test environments, and observation modalities as our agents. Five researchers with robotics expertise (but no prior ObjectNav experience) were given practice on the training scenes before evaluation. As shown in Table 4, they achieved near-perfect performance, with an average SR of 98%, consistently outperforming all existing autonomous methods. This highlights the substantial *algorithmic* gap between human and system performance, underscoring the need for more intelligent perception, reasoning, and exploration strategies to approach human-level robustness.

## 5 CONCLUSION

In this work, we present the first large-scale empirical study that systematically disentangles and evaluates the individual contributions of different components in RL-based Object-Goal Navigation systems. By decomposing these systems into three key modules: perception, policy, and test-time enhancement, we provide a clear and actionable understanding of what truly matters for performance. Our analysis reveals that perception quality and test-time enhancements are the most critical factors driving success, while improvements to the policy module alone yield marginal gains under current learning approaches. We demonstrate that simple plug-and-play strategies, such as dynamic goal selection and untrapping helpers, can significantly improve robustness without requiring retraining. Based on these findings, we propose practical design recommendations and develop a modular system that outperforms existing state-of-the-art methods on standard benchmarks. Moreover, our human baseline evaluation highlights the substantial performance gap between current agents and human-level navigation, pointing to future opportunities in closing this gap. We hope our study serves as a foundational reference for the community, guiding the design of more effective, interpretable, and generalizable ObjectNav systems, and inspiring future work toward principled benchmarking, modular learning, and real-world deployment.

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

## A  APPENDIX

The Appendix contains the following content:

- **LLM usage of the work** (AppendixA.1).
- **Reproducibility Statement** (AppendixA.2).
- **Failure case analysis** (AppendixA.3): In this section, we analyze failure cases across 1,000 test episodes to identify the primary error sources and their impact on object-goal navigation performance.
- **HM3D evaluation performance** (AppendixA.4)
- **Sensitivity of frontier-based method performance to sampling distance** (AppendixA.5)
- **Training details**(AppendixA.8): In this section, we provide details of the training setup used in our experiments.
- **Dynamic evaluation metrics**(AppendixA.9): In this section, we describe how the dynamic evaluation metrics are computed.
- **Perception module details**(AppendixA.10): In this section, we present the details of the perception module, including its structure, implementation, and integration within the overall navigation framework.
- **Policy module details**(AppendixA.11): In this section, we present the details of the policy module, including the observation space, architecture, action space, and reward.
- **Test-time enhancement module details**(AppendixA.12): In this section, we provide the details of the test-time enhancement module, along with the pseudo-algorithm illustrating how the strategy is applied during deployment.
- **Human expert baseline details**(AppendixA.13): In this section, we describe the human study conducted to evaluate RL agent performance against human experts.
- **Visualization of the perception module**(AppendixA.14): In this section, we provide visualizations of the perception module in Habitat-Sim.

## A.1 LLM USAGE

Large Language Models (LLMs) were used to aid the writing and polishing of the manuscript. Specifically, we used an LLM to assist in refining the grammar and language, improving the readability of the manuscript.

## A.2 REPRODUCIBILITY STATEMENT

We have made every effort to ensure that the results presented in this paper are reproducible. The codebase will be made publicly available to facilitate replication and verification.

## A.3 FAILURE CASE ANALYSIS

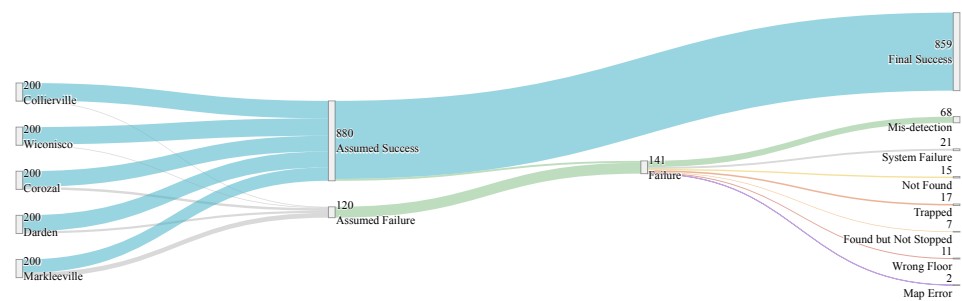

Figure 3: **Analysis of Test Navigation Scenarios**. Sankey plots illustrate the distribution of success and failure cases over 1,000 test episodes across five indoor scenes.

We analyze the test results over 1,000 episodes across five different test scenes, manually examining each case to identify failure points, with 200 episodes evaluated for each policy. From Figure 3, we identify 880 episodes as successful and 120 episodes as failures through manual inspection. However, 21 episodes were labeled as successful by human evaluation but classified as failures by the algorithm, due to the strict success criterion adopted in the Habitat simulator.

Among the remaining failure cases, 68 are attributed to mis-detections by the object detector, most commonly when a bed is incorrectly detected as a sofa. Of the remaining failures, 15 cases are caused by the agent not fully exploring the environment within the 500-step limit. 17 failures occur when the agent becomes trapped in narrow areas during navigation. In 7 cases, the agent continues exploring instead of stopping after locating the target object, typically due to inaccurate map projections in narrow spaces, which prevent the agent from reaching the target. 11 failures arise when the agent navigates to the wrong floor and subsequently loses the target object. Finally, 2 failures are attributed to map errors that prevent the agent from correctly localizing the target.

From the failure case analysis, we observe that most failures are caused by mis-detections, which further supports the view that the perception module plays a crucial role in determining the overall performance of object-goal navigation. Other failure cases, such as getting trapped during navigation or moving to the wrong floor, are largely related to the absence of test-time enhancement strategies. This also explains why such strategies play an important role in improving navigation performance.

## A.4 HM3D EVALUATION PERFORMANCE

To further assess the generality of our conclusions, we additionally evaluate our method on the HM3D indoor 3D scene dataset. We randomly sample five scenes from the HM3D test split, load them in Habitat, and measure navigation performance using the same configuration described in Section 4.1.

Table 5 reports the navigation performance across different perception modules, following the same comparison protocol as in Table 1. We use the rule-based corner-goal selection to isolate the effect of the perception module. The results on HM3D exhibit the same trend as in Gibson: stronger

Table 5: Evaluation of Perception Modules for Corner-Goal Policies on Five HM3D Test Scenes

| Det. | FPS | SR (%) | SPL (%) | D-SR (%) | D-SPL (%) |
|------|-----|--------|---------|----------|-----------|
| MRCNN | 21 | 47.8 | 21.3 | 37.6 | 20.2 |
| YOLO11-N | 25 | 47.4 | 18.9 | 31.4 | 17.2 |
| YOLO11-XL | 23 | 48.0 | 19.0 | 32.4 | 17.0 |
| RF-DETR-Seg | 22 | 52.4 | 24.2 | 39.6 | 22.4 |
| RF-DETR+SAM2 | 15 | 50.6 | 22.4 | 38.4 | 20.8 |

perception models yield better navigation performance. For instance, RF-DETR consistently outperforms MRCNN and YOLO. These findings further support our conclusion that perception accuracy correlates directly with overall navigation performance.

## A.5 SENSITIVITY OF FRONTIER-BASED METHOD PERFORMANCE TO SAMPLING DISTANCE

Previous work MOPA (Raychaudhuri et al., 2024) has examined how sensitive frontier-based exploration is to the frontier sampling distance, showing that exploration efficiency depends strongly on how far from a frontier the candidate goals are selected. This finding indicates an additional sub-component within the action-space module, namely candidate goal identification and selection, which can also influence overall performance. To further investigate this factor, we follow the idea introduced in MOPA and design three variants of goal-selection strategies: 1) Default (Yu et al., 2023a): selecting frontier goals within a predefined range and prioritizing frontier with the largest based on their area; 2) Nearest: selecting the frontier that is closest to the agent; 3) Farthest: selecting the frontier that is farthest from the agent. We conducted the experiment on our original Gibson test set with the RF-DETR-Seg detection modules, and measure the navigation performance.

Table 6: Comparison of Frontier Goal-Selection Strategies at Different Sampling Distances

| Det. | SR (%) | SPL (%) | D-SR (%) | D-SPL (%) |
|------|--------|---------|----------|-----------|
| Default | 72.9 | 40.3 | 53.6 | 36.6 |
| Nearest | 66.8 | 33.3 | 44.7 | 30.5 |
| Farthest | 58.7 | 36.9 | 43.9 | 30.3 |

From the experimental results in Table 6, we observe that, consistent with the findings reported in MOPA, the performance of frontier-based exploration is sensitive to the distance from the frontier at which the goal is sampled. Among the three variants we evaluated, the default selection strategy achieves the best performance, likely because it offers a good balance between information gain and travel cost. This preliminary result also suggests that when using discrete goals, an appropriate utility design for goal selection can contribute meaningfully to performance improvements. Selecting a suitable configuration may require some field tuning or prior knowledge.

## A.6 PPO CLIPPING PARAMETER STUDY

To study the effect of the learning algorithm's update constraint, we performed an ablation study on the PPO algorithm's clipping parameter, $\epsilon$. In our experiments, we considered two parameter settings: $\epsilon = 0.2$ and $\epsilon = 0.001$. We utilized our proposed policy framework and selected RF-DETR as the object detector. The resulting model was trained on the Gibson training dataset and evaluated on the Gibson evaluation dataset."

$\epsilon = 0.2$ is a common and widely adopted choice for various reinforcement learning task settings (Chaplot et al., 2020a; Zhang et al., 2023; Yu et al., 2023a; Xie et al., 2025).

For $\epsilon = 0.001$, setting the clipping parameter to such an extremely small value effectively shrinks the tolerance region of the old-to-new policy probability ratio to an exceptionally narrow range. This causes the PPO algorithm to behaviorally degenerate into an extremely conservative policy optimizer.

Table 7: Ablation Study of PPO Clipping Parameter

| Para. | SR (%) | SPL (%) | D-SR (%) | D-SPL (%) |
|---|---|---|---|---|
| 0.2 | 78.3 | 44.3 | 63.4 | 42.4 |
| 0.001 | 76.6 | 43.6 | 62.1 | 41.7 |

From Table 7, we observe that $\epsilon = 0.2$ consistently outperforms $\epsilon = 0.001$ across all evaluation metrics. This result indicates a central advantage of PPO: its clipped objective allows sufficiently large yet stable policy updates. When the clipping parameter is too small (e.g., $\epsilon = 0.001$), the update is over-restricted and learning stagnates, whereas a moderate value (e.g., $\epsilon = 0.2$) provides an effective balance between stability and learning efficiency.

### A.7 3D MAP REPRESENTATION EVALUATION

Recently, some works (Hong et al., 2023) use 3D voxel maps for object-goal navigation. Compared with 2D semantic maps, 3D voxel maps contain richer spatial information, which can potentially help the agent better understand the environment. However, the change in observation dimensionality from 2D to 3D will greatly increase the processing time of each environment interaction. RL approaches typically require millions of interactions to achieve good performance, making this increase particularly costly.

To further verify whether height information benefits navigation, we introduce an observation that includes additional height data. Specifically, we replace the standard binary obstacle map with a height map (Shah et al., 2025) based on the standard observation Xie et al. (2025) mentioned in the main text, where the height values are derived from the 3D point cloud during navigation. We still use RF-DTER as our object detector and employ Gibson as the environment for training and evaluation.

Table 8: Evaluating the Contribution of Height Information to Policy Performance

| Observation | SR (%) | SPL (%) | D-SR (%) | D-SPL (%) |
|---|---|---|---|---|
| With Obstacle Map | 78.3 | 44.3 | 63.4 | 42.4 |
| With Height Map | 75.1 | 43.3 | 63.1 | 41.9 |

The results in Table 8 do not show a significant improvement when switching from a binary obstacle map to a continuous height-map channel; in fact, the success rate decreases by roughly 3%. We interpret this outcome from two perspectives: (1) a continuous height map provides higher-dimensional information that is more difficult for the RL policy to encode and exploit effectively without a substantially larger and more diverse training set; and (2) Most of the ObjNav evaluations environments are single-floor-projecting the 3D structure into a 2D occupancy map already captures the information necessary for successful navigation. This abstraction yields a compact representation that aligns well with the task demands, which may explain why the height-map channel does not provide additional benefits within the same training and evaluation setup.

### A.8 TRAINING DETAILS

All experiments were conducted on multiple NVIDIA RTX 4090 GPUs. From the implementation details of the baseline methods, we observe that most prior works (Chaplot et al., 2020a; Yu et al., 2023a; Zhang et al., 2023; Xie et al., 2025) were trained for between 0.1 million and 10 million frames, with around 1 million frames typically yielding good performance. To balance performance and training cost, we trained all experiments for 2 million frames.

### A.9 DYNAMIC EVALUATION METRICS

Existing benchmarks for modular navigation methods often lack sufficient difficulty or fail to reflect the complexities of real-world environments. To address these limitations, we introduce a new evaluation setup, *Dynamic*, to better capture performance under more constrained and realistic settings:

Here is how to calculate the *Dynamic timestep*. In the previous benchmark setting, the maximum number of steps for one episode was fixed at 500, which means evaluation metrics were calculated either when the agent successfully reached the target or when it exhausted all 500 steps.

Now, we aim to restrict this maximum step count to a more reasonable and dynamic value. From the simulator, we can obtain the minimum distance from the agent's initial position to the target object. Based on this, we define the new maximum steps using the following formula:

$$\text{Max Dynamic Steps} = \alpha \times \left( \frac{D}{d} + \frac{360°}{\theta} \right)$$

where:

$D$: Initial distance to the target (in meters)

$d$: Agent movement distance per step (in meters)

$\theta$: Agent turning angle per step (in degrees)

$\alpha$: Scaling factor

With the newly defined dynamic maximum steps, we calculate the dynamic evaluation metrics (**D-SR**, **D-SPL**, and **D-DTS**). At the same time, we keep the maximum episode length fixed at 500 steps, under which we compute the standard metrics (**SR**, **SPL**, and **DTS**). In our setting, we choose $\alpha = 5$ to better distinguish the performance differences among various methods. If $\alpha$ is set too small, the evaluation becomes less discriminative, and all methods tend to exhibit similar results.

## A.10 PERCEPTION MODULE DETAILS

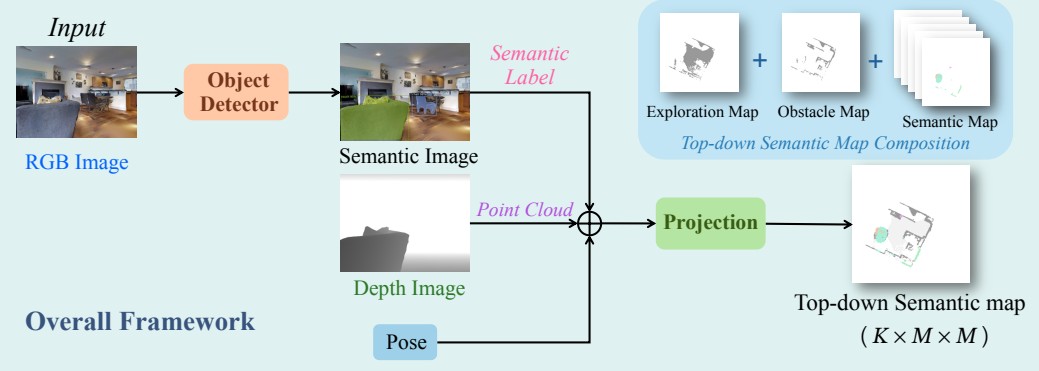

Figure 4: **Perception Module Overview.** RGB images are processed by a pretrained object detector for semantic labels, which are projected with depth-based point clouds to form a voxel map. Summing across height levels yields a multi-layer top-down semantic map, where $K$ is the channel number and $M$ the map size.

**Object Detector** The first step in the perception module is to extract semantic categories from the input RGB images. For the ObjectGoal navigation task, an effective object detector must not only be accurate but also capable of real-time inference. Although recent approaches such as the self-supervised model SAM (Kirillov et al., 2023; Ravi et al., 2024), and transformer-based models like Mask2Former (Cheng et al., 2022) and OneFormer (Jain et al., 2023), have demonstrated strong performance, CNN-based detectors remain the practical choice due to their efficiency on resource-constrained robots. Widely used models include Mask R-CNN (He et al., 2017), which processes only RGB inputs, and RedNet (Jiang et al., 2018), which leverages both RGB and depth inputs to improve accuracy.

**Map Generation** After obtaining the semantic segmentation image, semantic labels are projected onto a normalized 3D point cloud discretized into a voxel grid. Each occupied voxel is assigned a semantic category, resulting in a structured 3D semantic map.

To generate a 2D top-down map, voxel information is aggregated along the height axis. Summing all height levels produces an *exploration map*, while summing around the agent's height produces an *obstacle map*. The final semantic map is represented as a $K \times M \times M$ tensor, where $K = C + 2$: one channel each for the obstacle and exploration maps, plus $C$ channels for object categories. The map size $M$ controls the trade-off between spatial coverage, memory usage, and computation.

### A.11 POLICY MODULE DESIGN DETAILS

**Observation Space** In the perception module, we generate a top-down semantic map that encodes spatial and semantic information. Prior work such as SemExp (Chaplot et al., 2020a) and its successors (Yu et al., 2023a; Xie et al., 2025) use a high-dimensional $24 \times Map\ Size \times Map\ Size$ tensor as input, incorporating local/global maps and auxiliary features. However, this representation is inefficient because semantic channels are often sparse. Stubborn (Luo et al., 2022) showed that shuffling semantic channels had little effect, suggesting underutilization of these features.

To address this, we propose a compressed representation: compressing the semantic map into an RGB image, where obstacles, free space, and object categories are encoded via distinct colors. This preserves essential cues while reducing dimensionality and GPU memory usage.

**Action Space** For long-term goal planning, the action space determines how the agent selects a target position. Existing approaches generally fall into three categories: **continuous** (Chaplot et al., 2020b;a; Ramakrishnan et al., 2022; Zhai & Wang, 2023; Zhang et al., 2024), **discrete** (Zhang et al., 2023; Luo et al., 2022; Xie et al., 2025), and **frontier-based** (Yu et al., 2023a;b).

In the *continuous* setting, the policy outputs $(a_1, a_2) \in [0, 1]^2$, which are mapped to coordinates $(x, y)$. In the *discrete* setting, the agent selects a goal from a fixed set of candidates. In the *frontier-based* setting, the action space consists of frontiers, i.e., the boundaries between explored and unexplored regions, which are discretized using heuristics. The details are illustrated in Figure 5.

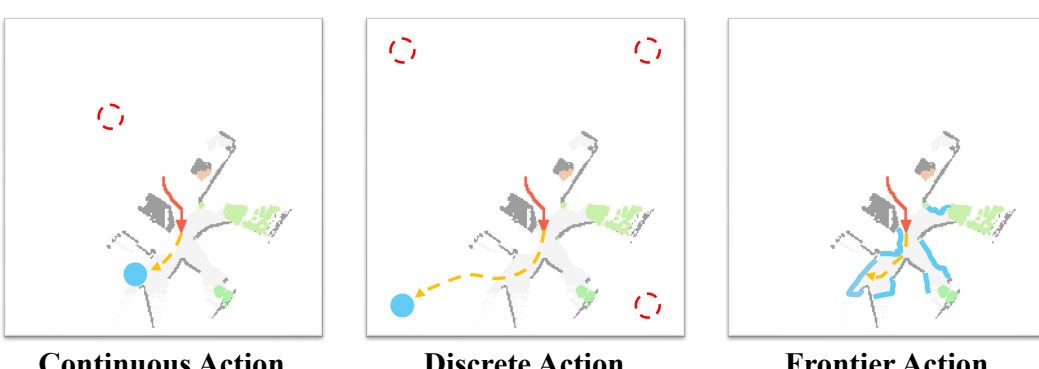

**Continuous Action**     **Discrete Action**     **Frontier Action**

Figure 5: **Action Space.** The current goal position is represented by a blue filled dot or line. For both *continuous* and *discrete* action spaces, red unfilled dots indicate the possible next goal positions. As shown in the map, in the *continuous* action space, the next goal can be located anywhere on the map. In contrast, for the *discrete* action space, the next goal is selected only from a predefined list of candidate positions. The yellow dashed line illustrates a potential navigation trajectory generated by the local policy.

**Network Architecture** For image-like inputs, CNNs remain a strong baseline (Chaplot et al., 2020a; Yu et al., 2023a; Zhang et al., 2023), while transformer-based models such as NaviFormer (Xie et al., 2025) have shown improved performance by treating maps as sequential data. In our setting, we consider standard pretrained backbones such as ResNet and ViT. For RGB-style semantic maps, the backbone is initialized with pretrained weights and fine-tuned during training. The detailed architecture is illustrated in Figure 6.

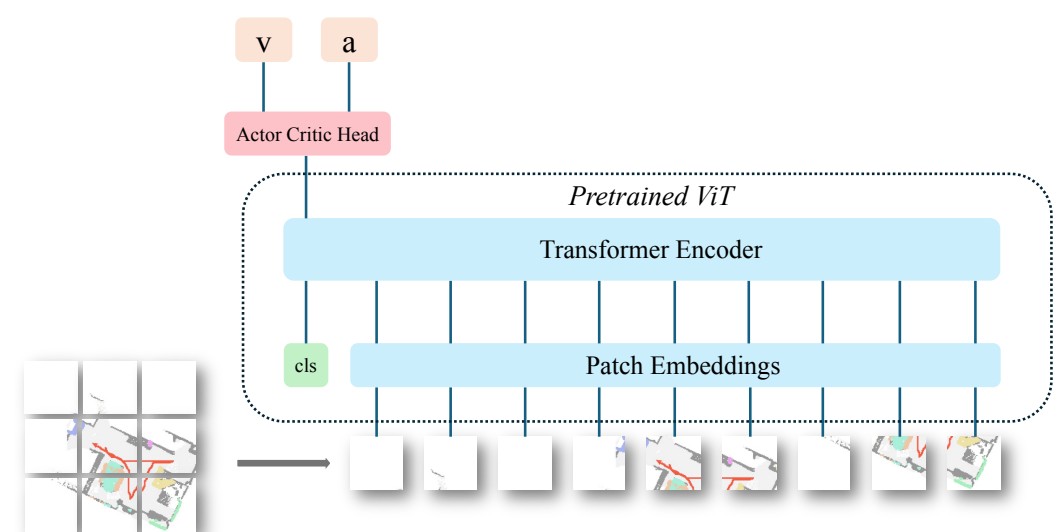

Figure 6: **ViT based Policy Network.** Our RL policy is adapted from a pretrained Vision Transformer (ViT). The compressed top-down semantic map is divided into 16 patches and passed through the transformer encoder, where the [CLS] token output is used to predict both action and value.

**Reward Design** For the ObjectGoal navigation task, we consider four types of rewards: exploration, distance-based, success bonus, and step penalty:

$$r_{\text{exploration}} = \alpha_1 \cdot (A_{\text{current}} - A_{\text{previous}})$$

$$r_{\text{distance to target}} = \alpha_2 \cdot (d_{\text{previous}} - d_{\text{current}})$$

$$r_{\text{success}} = \begin{cases} \alpha_3, & \text{if success} \\ 0, & \text{otherwise} \end{cases}$$

$$r_{\text{step penalty}} = -\alpha_4, \quad \text{every step}$$

where $A$ is the explored area, $d$ is the distance to the target, and $\alpha_1, \alpha_2, \alpha_3, \alpha_4$ are scalar weights.

### A.12 TEST-TIME ENHANCEMENT MODULE DETAILS

**Untrapped Helper** To address the issue of agents becoming trapped, previous work (Luo et al., 2022) introduced an untrapping helper strategy. During action execution, if repeated collisions exceed a predefined threshold, the agent activates the untrapping helper, which generates alternating turn-left and turn-right actions to escape the stuck situation. The details are provided in Algorithm 1.

---

**Algorithm 1:** Untrapping Helper

---

**Input:** Distance to obstacle $d$, collision threshold $\tau_{\text{coll}}$, block threshold $\tau_{\text{block}}$, previous action $a_{t-1}$

**Output:** Next action $a_t$

1 **if** $d < \tau_{coll}$ **then**
2     Increase blocked count by 1;
3 **end**
4 **if** *blocked count* $\geq \tau_{block}$ **then**
5     **if** $a_{t-1} = Move\ Forward$ **then**
6        $a_t \leftarrow$ Untrapping Helper's action (Turn Left or Turn Right);
7     **end**
8     **else**
9        $a_t \leftarrow Move\ Forward$;
10     **end**
11 **end**

---

To further demonstrate the effect of the untrapping helper visually, we provide an episode with the same starting point and the same object goal, comparing the results with and without the untrapping helper. As shown in Figure 7, the agent is prone to becoming trapped in confined areas such as a closet or other narrow spaces. With the untrapping helper, the agent is able to escape these situations, whereas without it, the agent ultimately fails to complete the episode.

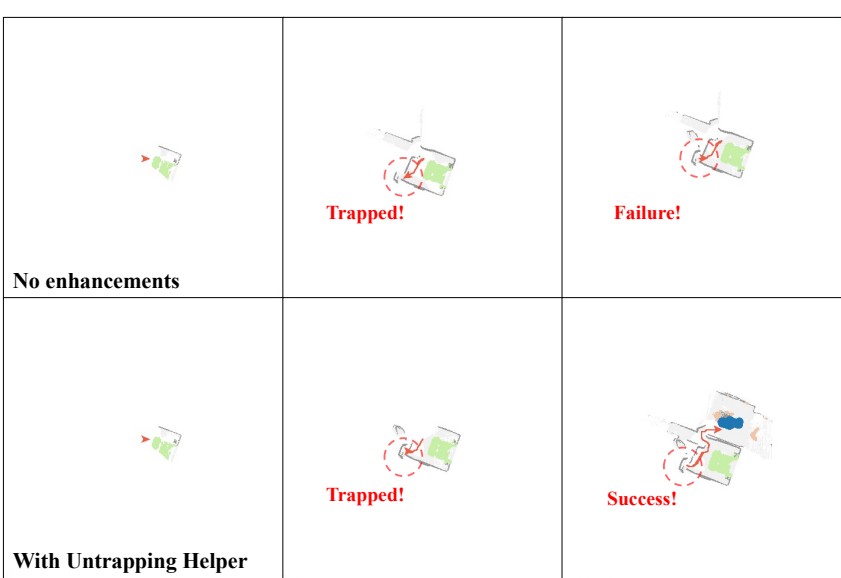

Figure 7: **Effect of the Untrapping Helper in Habitat.** Top row: without the untrapping helper. Bottom row: with the untrapping helper. Red dashed boxes highlight situations where the agent becomes trapped.

**Dynamic Goal Selection** Inspired by the performance of Stubborn (Luo et al., 2022), we extracted the dynamic goal selection heuristic from their algorithm and found that it can be effectively applied to RL-based policies to mitigate repeated exploration. Without dynamic goal selection, the long-term goal policy generates a new goal at a fixed frequency, which may cause premature abandonment of the current goal. We therefore designed a goal collector that stores the predicted goals and only switches to a new goal if the current goal is unreachable for an extended period or has been reached. The details are provided in Algorithm 2.

---

**Algorithm 2:** Dynamic Goal Selection

**Input:** Goal update frequency $f_{update}$, goal collector $g_{collector}$, distance to goal $d_{goal}$,
unreachable threshold $\tau_{\text{unreachable}}$, reached threshold $\tau_{\text{reached}}$
**Output:** Current goal $g_{current}$

1 **while** *training* **do**
2    **if** *step % $f_{update}$==0* **then**
3       $g_{collector} = RL\ policy\ prediction$
4    **end**
5    **if** $d_{goal} > \tau_{unreachable}$ *or* $d_{goal} < \tau_{reached}$ **then**
6       $g_{current} = g_{collector}$
7    **end**
8    $step + 1$
9 **end**

---

Here we also investigate the effect of dynamic goal selection qualitatively in Figure 8. We present an episode with the same starting point and the same object goal, comparing the results with and without dynamic goal selection. In complex indoor environments—such as those with multiple floors or large open areas—dynamic goal selection allows the agent to commit to exploring a single direction until reaching its end before switching goals. This prevents the agent from drifting into

other floors or performing redundant exploration. As shown in the example, with dynamic goal selection the agent explores one direction more consistently, whereas without it the agent changes direction more frequently and becomes trapped in the staircase.

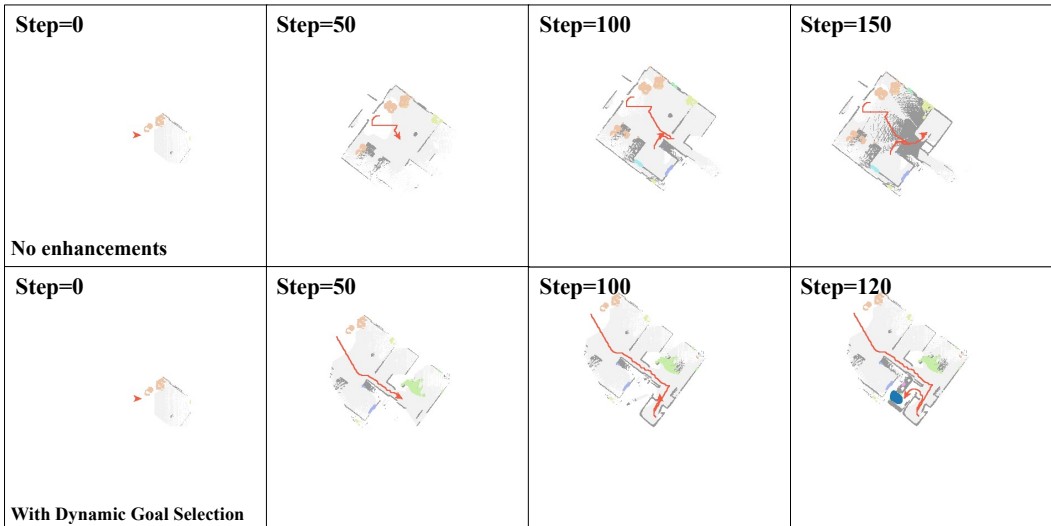

Figure 8: **Effect of the Dynamic Goal Selection in Habitat.** Top row: without the dynamic goal selection. Bottom row: with the dynamic goal selection.

**Remapping Mask** In the perception module, some prior work (Yu et al., 2023a;b; Xie et al., 2025) employs a *map augmentation* strategy to address the map-overlapping issue caused by staircases. Specifically, a special region on the map, referred to as the *remapping mask*, is defined. When the agent remains within this region for an extended number of steps—an indicator of ascending or descending a staircase—the current semantic map is cleared. A new top-down semantic map is then generated, preventing the accumulation of overlapping information across different floors.

Here we investigate the qualitative effect of the remapping mask in Figure 9. We illustrate one example to show how the remapping mask works. During navigation, although the task focuses on single-floor exploration, the agent may occasionally drift toward a staircase. Due to map projection, the staircase region is projected as an obstacle, causing the agent to become stuck on the stairs. With the remapping mask, once the agent detects that it has entered this masked region, it clears the current map and regenerates a new one, allowing it to resume normal navigation.

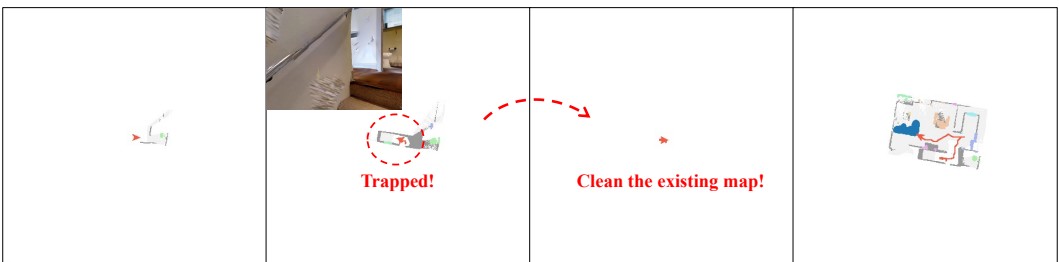

Figure 9: **Effect of the Remapping Mask in Habitat.**

## A.13 HUMAN EXPERT BASELINE DETAILS

To compare the RL agent against human performance, we conducted a controlled human study. Because humans can memorize indoor layouts after only a few trials, running all 1000 evaluation episodes is unnecessary and would not yield meaningful results. Instead, we selected a representative subset of test episodes such that the RL policy achieves nearly identical performance on the subset

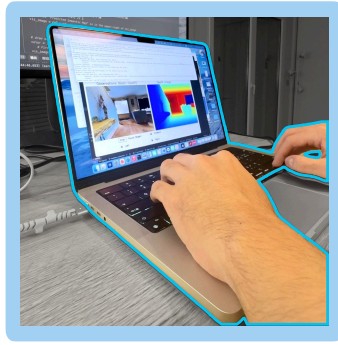 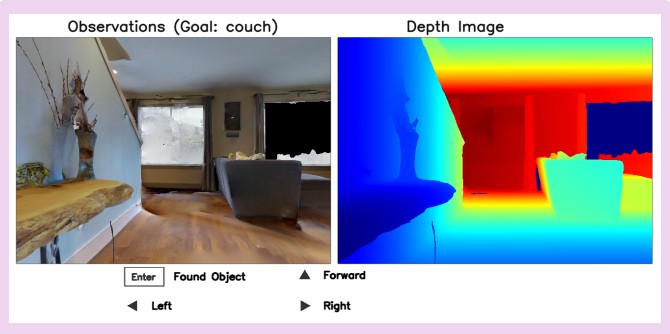

(a) Human Test Setup            (b) Input for Human Testers

Figure 10: (a) Overview of the human evaluation setup, including the control interface and interaction workflow. (b) The egocentric RGB-D observation and action interface shown to human participants during testing, identical to the inputs available to the RL agent.

and the full evaluation set. This ensures that the human comparison remains fair and statistically aligned with the agent's typical performance distribution.

Each participant is a robotics researcher with general navigation experience but no prior exposure to ObjectNav. The test setup is illustrated in Fig.10 (a). Before evaluation, participants complete a short practice session within the training environment used for the RL policy. This familiarization phase teaches the task rules, interface, sensor modalities, and action semantics, while avoiding exposure to test episodes. Once the participant demonstrates stable control of the agent and a full understanding of the task constraints, we begin the evaluation phase.

During evaluation, participants operate the agent using the same RGB-D egocentric view and discrete action space as the RL policy, as shown in Fig 10 (b). No additional global information, maps, or third-person views are provided. Each episode is executed independently, and participants are not allowed to repeat episodes or view prior trajectories, limiting potential memorization effects. In the supplementary material, we provide further details on the interface and evaluation protocol, including a screenshot of the user study UI.

### A.14 VISUALIZATION OF THE PERCEPTION MODULE

**Object detector.** For the same set of trajectories, we compare the performance of different object detector models within the Habitat simulator, as illustrated in Figure 11. The left column shows the original RGB input images, the middle column displays the output of the Mask R-CNN model without fine-tuning, and the right column presents the output of the Mask R-CNN model fine-tuned on the Gibson dataset. Each row corresponds to the same time step of the same trajectory.

As shown in the figure, the fine-tuned Mask R-CNN detects target objects more accurately than the non-fine-tuned model under identical conditions. For example, in the first row, it correctly identifies a chair even when only partially visible. In the second row, it successfully detects a toilet despite the significant distance. In the third row, it reliably identifies a potted plant, even with severe background distortion caused by simulator limitations.

**Map size.** To better understand the reasons behind the performance differences under varying map sizes, we provide visualizations of the top-down semantic maps in Habitat in Figure 12. For the same scene, a larger map corresponds to a larger spatial scale. Since the agent is reset to the center of the map every few steps when a new goal is predicted, the spatial interpretation of goal locations is affected by map size. In the case of the Corner Goal Policy, where the goal is always placed near the corners of the map, a larger map effectively sets more distant goals. This encourages the agent to explore a wider area, which can lead to improved navigation performance. This trend does not hold for the Frontier-Based Policy, where actions target frontiers of the explored space and are less sensitive to map size. In some cases, higher-resolution maps can degrade frontier extraction due to scaling effects, leading to worse performance, which explains why smaller maps sometimes perform better.

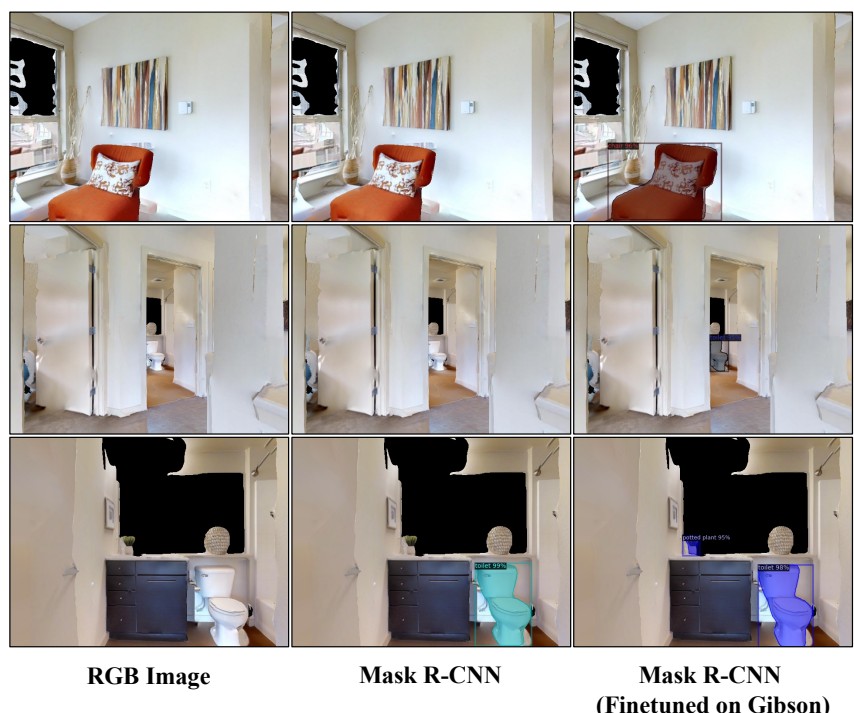

| RGB Image | Mask R-CNN | Mask R-CNN (Finetuned on Gibson) |

Figure 11: **Comparison of Different Object Detector Models in Habitat**

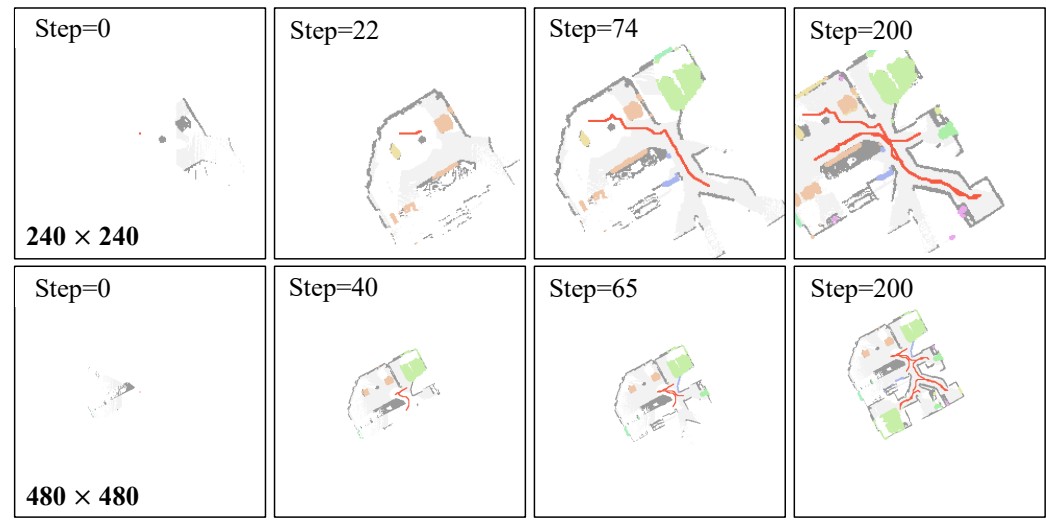

Figure 12: **Comparison of map sizes in Habitat.** Top row shows maps of $240 \times 240$. Bottom row shows $480 \times 480$.

**Map augmentation.** We also visualize the generated maps with and without augmentation under similar agent trajectories. As detailed in Figure 13, the overall trajectories are similar with and without map augmentation. However, the quality of the constructed maps differs noticeably. Without augmentation, scene details are often missing or inaccurate, while augmentation enables the agent to build richer and more complete maps within the same number of steps. This explains the performance gains under stricter evaluation. Under standard evaluation, agents have more time to explore, narrowing the performance gap. In the third column, we highlight a failure case: without augmentation, the agent misinterprets the staircase and gets stuck due to poor map quality.

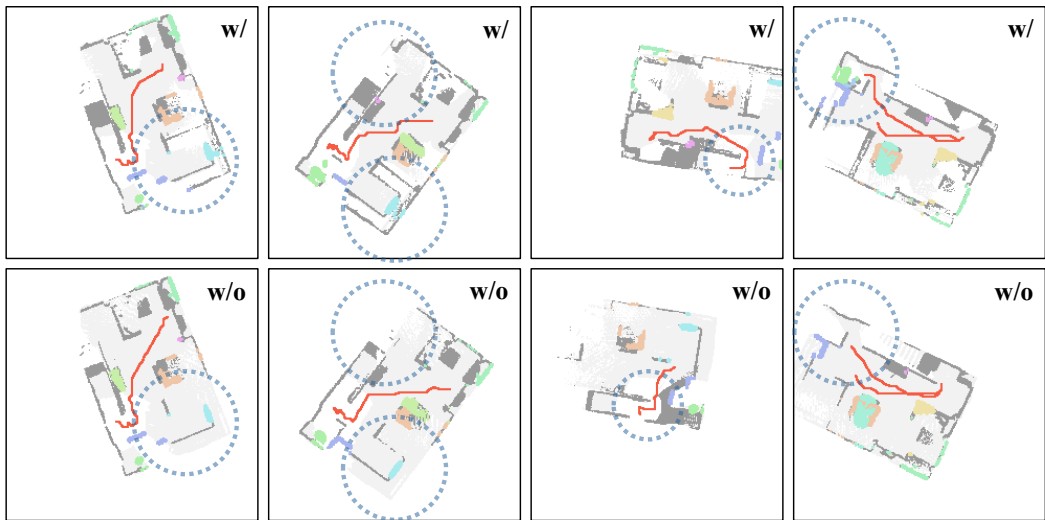

Figure 13: **Comparison of map augmentation in Habitat.** Top row: augmented maps. Bottom row: non-augmented. Blue dashed boxes highlight key differences.

