# OpenReview forum: "What Matters in RL-Based Methods for Object-Goal Navigation? An Empirical Study and A Unified Framework"
_ICLR.cc/2026/Conference — Submitted to ICLR 2026_

### Official Review · Reviewer_HZyA · 2025-10-30

**Soundness:** 3
**Presentation:** 2
**Contribution:** 2
**Rating:** 6
**Confidence:** 4

**Summary:**

This paper presents an empirical study to analyze the contributions of three different modules in RL-based Object-Goal Navigation systems: perception, policy and test-time enhancement. Among these, the authors find that perception and test-time enhancements are most crucial. Based on the findings, they propose design recommendations and develop a modular system that outperforms prior baselines.

**Strengths:**

The paper presents a diagnostic study on the contributions of different modules in RL-based ObjectNav systems, and provides useful insights that will benefit the community.

**Weaknesses:**

Overall, I find the study reported in the paper and the insights drawn to be useful for future research. However, I have some questions (listed below) and would request the authors to provide more clarification.

**Questions:**

1. Corner Goal Policy - The random exploration strategy seems similar to MOPA[1], but the citation is missing.
2. Frontier-Based Policy -
    a) There could be different ways to sample the frontier goal from the list of frontiers, such as selecting the one nearest to or farthest from the agent, etc. Did you try these variants?
    b) Did you try semantic frontier selection like VLFM[2]?
    c) Referring to MOPA, they find that the performance is sensitive to the distance at which the frontier is sampled. Did you observe similar behavior?
3. Object detector - Why didn’t you try more recent detectors (Grounding DINO[3])?
4. L260-261 “global map is only used to update the local map and is not directly used for navigation” - can you elaborate what you mean by this? I believe the global map is used to explore the environment and to memorize past observations. The local map only gives a limited view of the area and will not be sufficient in long-horizon task planning.
5. Map augmentation - can you elaborate what you mean by augmentation and what different techniques you tried?
6. Which ObjectNav dataset did you use for your experiments?

[1] Raychaudhuri et al. MOPA: Modular Object Navigation with PointGoal Agents. 2024.
[2] Yokoyama et al. VLFM: Vision-Language Frontier Maps for Zero-Shot Semantic Navigation. 2023.
[3] Liu et al. Grounding DINO: Marrying DINO with Grounded Pre-Training for Open-Set Object Detection. 2024.

---

> ### Author Response · Authors · 2025-11-21
> **Official Comment by Authors (1/2)**
>
> Thank you for the insightful review and questions. We have submitted a revised version addressing your comments. Please find our detailed responses below:
>
> >Q1: Corner Goal Policy - The random exploration strategy seems similar to MOPA [1], but the citation is missing.
>
> Thank you for pointing out this missing citation. MOPA [1] indeed introduces randomness when selecting navigation goals from detected candidates (e.g., frontiers), which is related to our goal-selection strategy. Its findings are consistent with ours: incorporating randomness into goal selection can improve overall exploration efficiency. We have now added the appropriate citation to MOPA, and added additional experiments inspired by it. We appreciate the reviewer bringing this to our attention.
>
> >Q2: Frontier-Based Policy - a) There could be different ways to sample the frontier goal from the list of frontiers, such as selecting the one nearest to or farthest from the agent, etc. Did you try these variants? b) Did you try semantic frontier selection like VLFM[2]? c) Referring to MOPA, they find that the performance is sensitive to the distance at which the frontier is sampled. Did you observe similar behavior?
>
> a) While we agree that there are many possible variants for selecting a frontier goal (e.g., nearest, farthest, or other ranking criteria based on different utilities), our intention in this work was to provide a broad-level analysis of different categories of goal selection mechanisms.
>
> However, we do agree that studying these different variants could also be valuable, and we now provide an anonymous GitHub repository and plan to include a compact, flexible configuration that allows switching between different goal-selection strategies. This will make it straightforward to add and test these variants in the future.
>
> b) Semantic frontier selection methods such as VLFM [2] require substantial preprocessing and frequent map updates based on perception inputs. In our RL training setup, this would require running large, computation-heavy vision models (e.g., BLIP in VLFM) at each step, which makes the training process infeasible. For this reason, such semantic frontier approaches were not evaluated in our pipeline.
>
> c) We have added an additional experiment of different goal selection approaches inspired by MOPA. Our observations are consistent with the findings reported in MOPA, which show that frontier-based exploration performance is highly sensitive to the distance from the frontier at which goals are sampled. The detailed results are presented in Table I and also included in A.5 of our supplementary material. We explicitly design three variants of the frontier-based baseline:
>
> - sampling frontier goals only within a predefined distance range (i.e., neither too close nor too far from the agent) and prioritizing frontiers by their area;
>
> - always selecting the closest frontier;
>
> - always selecting the farthest frontier.
>
> **Table I: Comparison of Frontier Goal-Selection Strategies at Different Sampling Distances**
>
> | Det.     | SR (%) | SPL (%) | D-SR (%) | D-SPL (%) |
> |----------|--------|---------|----------|-----------|
> | Default  | 72.9   | 40.3    | 53.6     | 36.6      |
> | Nearest  | 66.8   | 33.3    | 44.7     | 30.5      |
> | Farthest | 58.7   | 36.9    | 43.9     | 30.3      |
>
> The findings from this additional focused test highlight the strong influence of goal-selection strategy on final navigation performance. We believe that a more general conclusion could be drawn through a compact set of follow-up experiments that systematically explore goal-sampling configurations to identify robust and effective choices. We will also integrate this flexibility into our open-sourced repository so that users can easily tune and test different strategies. We sincerely appreciate the reviewer for pointing out this important factor, as it broadens the scope of our study and makes our conclusions more useful for the system design of future work.
>
> >Q3: Object detector - Why didn’t you try more recent detectors (Grounding DINO [3])?
>
> Thank you for the helpful suggestion. As mentioned in our responses to other reviewers, our initial goal was to benchmark the most commonly used perception models in the literature. That said, evaluating more recent detectors is indeed valuable. In the revised version, we have added several modern detectors—YOLOv11, RF-DETR, and SAM2—and the corresponding results are reported in Table 1 and Table 5.
>
> We also experimented with large open-vocabulary detectors such as GroundingDINO. However, obtaining bounding boxes requires multiple inference rounds per query, which reduces the throughput to around 1 Hz, making it impractical for both inference and RL training in our setting.
>
> [1] MOPA: Modular Object Navigation with PointGoal Agents
>
> [2] VLFM: Vision-Language Frontier Maps for Zero-Shot Semantic Navigation
>
> [3] Grounding DINO: Marrying DINO with Grounded Pre-Training for Open-Set Object Detection

---

> ### Author Response · Authors · 2025-11-21
> **Official Comment by Authors (2/2)**
>
> >Q4: L260-261 “global map is only used to update the local map and is not directly used for navigation” - can you elaborate what you mean by this? I believe the global map is used to explore the environment and to memorize past observations. The local map only gives a limited view of the area and will not be sufficient in long-horizon task planning.
>
> Thank you for pointing this out. In this context, “navigation’’ specifically refers to the low-level planning procedure that the agent uses to reach the current goal. This low-level planner operates only on the local map, rather than directly on the global map.
>
> At every step, the agent receives an RGB-D observation, from which it updates its local map. At a fixed interval, the updated local map is stored in the global map, which indeed serves as the memory of past observations and supports long-horizon exploration. Afterward, the agent uses its current estimated pose to retrieve the relevant region from the global map to refresh its local map.
>
> Our intention in this section is to study how the perceptual scale of the agent (i.e., the size of the local map it can directly operate on) influences the navigation performance. The global map is used for long-term memory and exploration,  but the action-level, low-level planning is executed solely on the local map.
>
> >Q5: Map augmentation - can you elaborate what you mean by augmentation and what different techniques you tried?
>
> Our “map augmentation’’ refers to a set of fine-grained enhancements applied during the 3D-to-2D projection stage. Here, we summarize these enhancements as follows:
>
> (1) integrating voxel features across multiple height bands, which alleviates the issue of missing obstacles or stairs when relying on a single height slice;
>
> (2) correcting inconsistent obstacle/free-space predictions that often arise from depth noise in the absence of augmentation;
>
> (3) applying local spatial masking to suppress spurious stair-like patterns that would otherwise leak into unrelated regions of the map. Together, these augmentations yield a top-down map that is more robust to depth errors and more faithfully aligned with the real scene.
> >Q6: Which ObjectNav dataset did you use for your experiments?
>
> We used the Gibson dataset in Habitat for all experiments in the original submission. In the revised version, we additionally include experiments on the HM3D dataset to further evaluate the generality of our conclusions. Details of both setups are provided in the revised version.

---

> ### Author Response · Authors · 2025-11-27
> **A Gentle Reminder**
>
> Dear reviewer HZyA, thank you once again for your time and thoughtful review. We sincerely appreciate your feedback!
>
> The end of the discussion period is rapidly approaching, and we would greatly appreciate it if you could review our response and let us know whether your concerns are adequately addressed. If not, or in case you have any further concerns, we would be more than happy to work with you on improving the paper.

---

### Official Review · Reviewer_ehQz · 2025-10-31

**Soundness:** 3
**Presentation:** 3
**Contribution:** 2
**Rating:** 6
**Confidence:** 4

**Summary:**

The paper presents a systematic empirical study of ObjectNav systems focusing on RL-based modular approaches found in the existing literature. The paper studies individual contributions of various existing subsystems (perception, policy, and test-time enhancement) in controlled experiments, and presents detailed analyses and recommendations. The paper integrates these insights to develop an ObjectNav approach that outperforms existing approaches, and also highlights the current gap with human-level performance in these settings.

**Strengths:**

- The paper is well-written, with clearly defined goals and scope that is relevant to the community.
- The paper unifies the existing literature in the areas of modular RL-based ObjectNav approaches with clear and valuable insights into the individual modules, along with recommendations for researchers aiming to deploy or improve these systems.
- The experiments are comprehensive and well-designed.

**Weaknesses:**

- Many findings in the paper reinforce existing design choices well-known to the community. There are no novel paradigms explored in the paper, neither does the paper present any theoretical contributions. Yet, I believe the paper will be useful to the community as it grounds our intuitive design choices though empirical validation.
- The experiments on the choices of observation spaces, action spaces and network architectures with regards to the policy module feels limited, especially since it considers the policy as a black-box. There are policies in ObjectNav settings that use hierarchical or model-based approaches, which are not considered in the paper but are quite relevant and important to consider. Consequently, the claim “policy improvements with current methods yield only marginal gains” feels a little too strong in this regard.

**Questions:**

- With regards to my above concern about limited experiments with policy module, why were other policy learning approaches not considered? I think scoping the rationale behind this would be valuable to the readers.
- Minor comment: Please put citations inside parentheses when they are not a part of the sentence (see the use of `\citet` and `\citep` and use them appropriately).

---

> ### Author Response · Authors · 2025-11-24
> **Official Comment by Authors (1/2)**
>
> >W1: There are no novel paradigms explored in the paper, neither does the paper present any theoretical contributions. Yet, I believe the paper will be useful to the community as it grounds our intuitive design choices though empirical validation.
>
> Thank you for acknowledging the usefulness of our contribution. We agree with your assessment. This intuition was part of our motivation for conducting a large-scale, systematic study to formally evaluate and ground common design choices in ObjectNav systems.
>
> Our goal is to contribute to the community by highlighting where performance gains genuinely come from and by identifying directions that can further advance ObjectNav. While many recent works focus on increasingly complex policy architectures, we found that the field still lacks a standardized evaluation framework. Our study fills this gap by providing a clear, modular formulation and by examining each component in detail. In addition, our experiments reveal several new and previously unreported findings—for example, the substantial impact of test-time enhancement, which is widely used in recent works but rarely analyzed systematically, and the large performance gap between our trained state-of-the-art system and human experts, suggesting significant room for progress in policy learning and scene understanding.
>
> Overall, we see our main contribution as: 1) summarizing and formalizing the modularized ObjectNav system design, 2) establishing a standardized development and evaluation pipeline, and 3) providing constructive insights to guide future research in this domain.
>
> We also plan to support the community by open-sourcing our framework. Our initial anonymous GitHub repository is now available:  https://anonymous.4open.science/r/What-matters-in-RL-ObjNav-84A3
>
> >W2: The experiments on the choices of observation spaces, action spaces and network architectures with regards to the policy module feels limited, especially since it considers the policy as a black-box. There are policies in ObjectNav settings that use hierarchical or model-based approaches, which are not considered in the paper but are quite relevant and important to consider. Consequently, the claim “policy improvements with current methods yield only marginal gains” feels a little too strong in this regard.
>
> Thank you for the insightful comments. First, we do consider model-based approaches in our evaluation. Both the **Corner Goal Policy** and the **Frontier-Based Policy** in Section 4.1 are fully model-based. In the revision, we have also added several additional variants of the frontier-based strategy to study this class of methods more thoroughly (A.5). We agree that understanding the gap between rule-based and learning-based approaches is important, which is why we explicitly evaluate non-learning goal-selection methods in this section.
>
> Second, our setup is inherently **hierarchical**: the policy outputs high-level goals, while low-level control and mapping are handled by separate modules. We intentionally avoid treating the entire ObjectNav policy as a black box, as such monolithic designs have been shown to be less robust in sim-to-real transfer [1]. Our modular formulation increases transparency and allows us to study action spaces, goal-selection strategies, perception modules, and mapping components independently.
>
> We agree that there is still room for improving policy architectures—for example, through careful hyperparameter tuning or more sophisticated network designs. Our claim that “policy improvements with current methods yield only marginal gains” is meant in the context of our standardized pipeline: given strong perception and mapping, the incremental benefit from policy-only modifications was consistently smaller compared to improvements from upstream components. Our goal is to highlight that substantial and easily overlooked gains can be achieved by improving other modules before focusing on fine-grained policy design, which is an aspect often missing in current literature.
>
> Finally, we emphasize that our framework is designed to support future work on low-level policy improvements, hierarchical designs, and alternative architectures. We hope the community will benefit from a more standardized and unified pipeline for developing and evaluating ObjectNav systems.
>
> [1] Navigating to objects in the real world

---

> > ### Author Response · Authors · 2025-11-24
> > **Official Comment by Authors (2/2)**
> >
> > >Q1: With regards to my above concern about limited experiments with policy module, why were other policy learning approaches not considered? I think scoping the rationale behind this would be valuable to the readers.
> >
> > Thank you for the question. We base our policy module on PPO because nearly all existing RL-based ObjNav works adopt PPO or PPO-style on-policy RL as the standard backbone. This is largely due to PPO’s stability under the sparse and delayed rewards characteristic of ObjectNav, its robustness in long-horizon navigation tasks, and its consistently strong empirical performance in Habitat-based benchmarks. Our experiments further confirm that PPO provides reliable and stable convergence. This observation is consistent with broader trends in robotics RL, where PPO is the dominant choice across quadrupedal locomotion [2][3], aerial navigation [4][5], and humanoid control [6], due to its reliability and ease of tuning. Given this landscape, we selected PPO-based policies to ensure comparability with prior work and to maintain clean, controlled cross-module analysis.  We appreciate the reviewer’s suggestion and have incorporated this justification and the new results into the revised manuscript.
> >
> > At the same time, we have now open-sourced the full framework in an anonymous GitHub repository: https://anonymous.4open.science/r/What-matters-in-RL-ObjNav-84A3; we warmly encourage the broader community to explore and evaluate such alternative policy designs within our modular setup. The framework is structured to make these extensions straightforward and reproducible.
> >
> > In addition, the reviewer’s comment motivated us to include some ablation studies on the effect of PPO’s parameter to demonstrate how “policy learning algorithms can affect the performance”. While we will keep updating the results during the rebuttal phase, we now include a comparison of the policy gradient clipping factor (Table 3 in the revised manuscript). We find that the standard setting performs clearly better, while the smaller value severely restricts learning.
> >
> > >Q2: Minor comment: Please put citations inside parentheses when they are not a part of the sentence (see the use of \citet and \citep and use them appropriately).
> >
> > Thank you for the helpful suggestion. This was a great catch, and we have updated the citation formatting accordingly in the revised manuscript.
> >
> > [2] Learning agile and dynamic motor skills for legged robots
> >
> > [3] Learning quadrupedal locomotion over challenging terrain
> >
> > [4] Champion-level drone racing using deep reinforcement learning
> >
> > [5] Multi-Task Reinforcement Learning for Quadrotors
> >
> > [6] Real-World Humanoid Locomotion with Reinforcement Learning

---

> > > ### Author Response · Authors · 2025-11-27
> > > **A Gentle Reminder**
> > >
> > > Dear reviewer ehQz, thank you once again for your time and thoughtful review. We sincerely appreciate your feedback!
> > >
> > > The end of the discussion period is rapidly approaching, and we would greatly appreciate it if you could review our response and let us know whether your concerns are adequately addressed. If not, or in case you have any further concerns, we would be more than happy to work with you on improving the paper.

---

### Official Review · Reviewer_xY4S · 2025-11-07

**Soundness:** 2
**Presentation:** 2
**Contribution:** 1
**Rating:** 2
**Confidence:** 5

**Summary:**

Prologue: ObjectGoal task has a large flood of papers. It started with pre-trained models and over-fitting to dataset and environment to claim prowess over SOTA and gradually the shift has happened towards prowess of LLM/MLM used when zero shot approaches ourtperform earlier pre-trained models (RL wrt env. rewards or supervised wrt VLA).

The paper:
This is an empirical study (expanded ablation) for RL agents' performance in Sim for ObjNav task. This is an engineering paper more than a research paper, but with commendable hacks/tweaks. The authors divide ObjectGoal into 3 stages - Perception (scene graph / map), Policy (RL agent), and Test-time Enhancement (rule-based heuristics / hacks). The main finding is the first module, Perception being a decider in final success, although this is expected as initial errors add to latter as cascade.

Epilogue: Hard effort has been put in for this work, however, after a few days I am pretty sure a claimed new but old-tweaked method will come proving better results than this - that is the speed of research community in this sub-domain. Lack of codebase and theory makes it hard to verify claims which are just numbers. As such for ICLR community, this does not add any value wrt learning representations for Object Goal task, which is already not known. Appreciating the work, requesting authors to focus on representation part of Object Goal -- which can be better than how a MLM represents the scene perception internally.

**Strengths:**

1. Extensive experiments on RL SOTA on ObjectNav task.
2. In contrast to other papers, that omit critical workarounds, this paper gives detailed logical accounts of ways to engineer better performance.
3. Section A.5 list down a modified way to evaluate, which is practical; however the SOTA probably tried their best wrt current norm of 500 step counts.
4. Fig. 3 partially presents the Failure case analysis wrt what exactly caused the failure - something commendable -- however this is too environment specific to claim generalization comments and decisions in algo/RL reward design.

**Weaknesses:**

1. Modular aspects of Embodied AI Goal tasks are already covered by previous works -- the philosophy is already proven. Reference: Wu, Qiaoyun, et al. "Image-goal navigation in complex environments via modular learning." IEEE Robotics and Automation Letters 7.3 (2022): 6902-6909. And CSR that shows just by changing representation, better results can be fetched in downstream tasks -- Gadre, Samir Yitzhak, et al. "Continuous scene representations for embodied ai." Proceedings of the IEEE/CVF conference on computer vision and pattern recognition. 2022.
2. The policies and architectures tested are straightforward with no discussion wrt their selection choice and failure case analysis.
3. There is lack of qualitative results to support the claims.

**Questions:**

1. In perception, why DINO + SAMg or OWLVT2 is not used which works pretty well for open vocabulary objects now in indoor scenes? MaskRCNN is outdated in current context.
2. Line 464 is not believable - in terms of human experts. Firstly, 5 experts do not add statistical significance. Secondly, human users with RGB-D ego view only (not 3rd part FPS type view) cannot navigate complex paths as smoothly (action space) as a pre-trained agent in similar environment. Requesting repeat of experiments in ego view only.
3. In table 3, there is sudden jump in success for applying heuristics for obstacle avoidance helper.
4. Line 941: How can agent come out if it is stuck at a corner and is rotating in a loop when dynamic goals swap by oscillation?
5. Instead of quite old Sem-Exp type map, probably two extreme ends could be tested - like simple semantic 3D voxel map and 3DLLM map [https://arxiv.org/pdf/2307.12981] to detect the prowess of map representation in downstream tasks and overhead.

---

> ### Author Response · Authors · 2025-11-24
> **Official Comment by Author (1/4)**
>
> We respectfully disagree on the term that *“As such for ICLR community, this **does not add any value** wrt learning representations for Object Goal task, which is already not known."*
>
> Our study is indeed not intended as a representation-learning paper, and we do not claim to introduce new representation learning techniques. Instead, our contribution lies in systematically analyzing how different modules in the ObjNav pipeline contribute to end-task performance. This addresses a clear gap in the literature. Despite the many ObjNav papers published in recent years and the fast advancing speed mentioned by the reviewer, the field still lacks a principled, cross-module understanding of where performance gains actually originate. And we believe that providing this structured, holistic analysis is a meaningful and nontrivial contribution.
>
> It is also worth noting that the other reviewers explicitly acknowledge the usefulness of our study and the value it provides to the community. Reviewer eFcx noted that our findings were “very interesting” and that the inclusion of human baselines was “valuable.” Reviewer ehQz stated that our work is “useful to the community as it grounds our intuitive design choices through empirical validation.” Reviewer HZyA commented that “the study reported in the paper and the insights drawn are useful for future research.” Their assessments align with our motivation: to provide strong empirical grounding, clarify which design choices have a meaningful impact on performance, and offer a transparent framework that supports future methodological progress. We also appreciate that you acknowledge our hard work, which we also aim at having better interpretation and understanding of the progress in this field, to better analyse and compare the performance in the literature, as the reviewer mentioned, we wanted to contribute both framework and codebase to have fair evaluate and verification of the numbers.
>
> However, we are genuinely thankful for the reviewer’s suggestion to clarify and better position the contributions of our work. In response, we have revised the contribution section in the manuscript to clarify our goals and highlight the scope of our study. To further demonstrate our commitment to providing a useful and transparent framework for the ObjectNav community, we have already released an anonymous GitHub repository containing the full framework, as well as the codebase for human evaluation. We encourage the broader community to utilize these resources to verify our findings and contribute to advancing research in this area.
>
> Given these revisions and the additional transparency provided through the released resources, we hope the reviewer may reconsider their assessment of the contribution and its relevance to the ICLR community.

---

> ### Author Response · Authors · 2025-11-24
> **Official Comment by Author (2/4)**
>
> >W1:Modular aspects of Embodied AI Goal tasks are already covered by previous works -- the philosophy is already proven. Reference: Wu, Qiaoyun, et al. "Image-goal navigation in complex environments via modular learning." IEEE Robotics and Automation Letters 7.3 (2022): 6902-6909. And CSR that shows just by changing representation, better results can be fetched in downstream tasks -- Gadre, Samir Yitzhak, et al. "Continuous scene representations for embodied ai." Proceedings of the IEEE/CVF conference on computer vision and pattern recognition. 2022.
>
> Thank you for the comment. We agree that modular learning is an established paradigm, and we do not claim conceptual novelty in adopting a modular structure. The works mentioned in this review, however, each focus on a single module. Wu et al. [1]  concentrate on the goal prediction module, while Gadre et al. focus on scene representation.
>
>  In contrast, our contribution is a unified, controlled analysis across all major modules of the ObjNav pipeline: perception, policy, and test-time enhancement. To our knowledge, prior work does not provide a systematic, cross-module examination of where performance gains actually originate, nor does it quantify the relative contributions of each module under a shared evaluation protocol.
>
> At the same time, we are aware of these works, but we did not include them as direct baselines because neither is evaluated on the ObjectNav task. Reference [1] focuses on ImageNav, where navigation is conditioned on a target image rather than an object category, and reference [2] studies a room-rearrangement task. Their analyses also remain within a single component of the pipeline.
>
> However, we believe including these works could improve the coverage of existing literature, and we have included both in the revised manuscript.
>
> [1] Image-goal navigation in complex environments via modular learning. IEEE Robotics and Automation Letters 7.3 (2022): 6902-6909.
> [2] Continuous scene representations for embodied AI. Proceedings of the IEEE/CVF conference on computer vision and pattern recognition. 2022.
>
> >W2: The policies and architectures tested are straightforward with no discussion wrt their selection choice and failure case analysis.
>
> We had a failure case study(A.1 in supplementary). Regarding the policy and architecture design, we focus on general-level choices, such as CNN vs. Transformer, different action spaces, etc. We did notice that there are more sophisticated lower-level designs for the model architectures, but we focused on giving generic advice. As we found in our final study results, the lower-level architecture design does not have as significant an influence on final performance as the other components (perception, test-time enhancement, etc.), as you can observe from our results with several baseline methods (Table 4). We would also like to point out that our framework will enable further study of different lower-level architecture designs and their evaluation in a more scientific manner.
>
> >W3: There is lack of qualitative results to support the claims.
>
> Thank you for the suggestion regarding qualitative results. We are not fully certain which specific claims the reviewer feels lack qualitative support. In the original submission, we did include qualitative visualizations, such as comparisons across different detection modules, action spaces, and map sizes. We placed them in the appendix due to page limits.
> Following your suggestion, we have added additional qualitative results in the revised appendix. In particular, for each of our test-time enhancement modules, we now include representative visualizations showing how exploration trajectories improve when the enhancement is applied. These examples provide direct evidence of the behaviors we discuss and illustrate the circumstances under which each enhancement is most beneficial.
> We are happy to include additional qualitative results if needed. We would also like to note that our claims are primarily supported by extensive quantitative evaluations. Because the improvements we report reflect overall averaged gains, they are often more clearly captured through quantitative metrics than through isolated qualitative snapshots. For this reason, we believe the quantitative evidence offers a stronger and more reliable basis than a small set of qualitative examples. We also appreciate that the other reviewers recognized the value of our empirical analysis. Nevertheless, our framework makes it easy to generate qualitative visualizations, such as robot trajectories, from any trained checkpoints. This ensures the users can readily produce more examples upon request.

---

> ### Author Response · Authors · 2025-11-24
> **Official Comment by Author (3/4)**
>
> >Q1: In perception, why DINO + SAMg or OWLVT2 is not used which works pretty well for open vocabulary objects now in indoor scenes? MaskRCNN is outdated in current context.
>
> Thank you for this insightful suggestion. As noted in our responses to other reviewers (eFcx and HZyA), we have expanded our perception experiments in the revised version. Specifically, we now include several more recent object detectors, including YOLOv11, RF-DETR, and SAM2, with results reported in Table 1 and Section A.4. These additions strengthen our findings and confirm our initial conclusion: stronger perception generally leads to better navigation performance. We appreciate the reviewer’s comment for helping us make this section more complete and rigorous. We also observe that our fine-tuned Mask R-CNN still achieves the best performance overall. This suggests that targeted fine-tuning on the task domain remains highly beneficial, likely because the model becomes particularly effective at handling challenging Habitat-specific cases (e.g., examples shown in Fig. 11). This observation complements our main conclusion that improving in-domain perception accuracy is beneficial.
> Regarding open-vocabulary detectors, we did evaluate an open-vocabulary setup using GroundingDINO, as mentioned in our reply to reviewer HZyA. However, building a semantic map from open-vocabulary predictions requires running the detector multiple times per frame (one per queried category), and using a fixed object list prevents the system from being truly “open vocabulary.” More importantly, this process reduces the system’s throughput to below 1 Hz, making RL training infeasible in our setting.
> An alternative is to use language-embedded feature maps, as in VLFM[1], but this approach requires substantial computation and memory overhead during high-dimensional map construction, which again makes RL training extremely challenging due to the high memory consumption and makes each interaction with the environment (which RL methods heavily rely on) extremely slow.
> We acknowledge the potential of this direction and agree that integrating open-vocabulary representations into ObjectNav is an exciting research opportunity. We would be very interested in suggestions or promising directions from the reviewer or the community. However, this would require more future research on more memory and computationally efficient representation, and at the moment, beyond the scope of our study.
>
> [1] VLFM: Vision-Language Frontier Maps for Zero-Shot Semantic Navigation
>
> >Q2: Line 464 is not believable - in terms of human experts. Firstly, 5 experts do not add statistical significance.
>
> Thank you for the comment. Our goal with the human expert baseline is not to make fine-grained psychometric claims, but to assess whether current ObjNav benchmarks are saturated by reasonably skilled users. So it is more important for us to showcase that there exist better human experts that can outperform SotA algorithms, rather than making the claim that “average human” are better than the SotA algorithms.
>
> Meanwhile, we would also clarify that we have 50 evaluations from the five experts across different scenes in total. More importantly, all experts consistently outperform our best RL agent by a clear margin that is much larger than the variance across participants, making the qualitative conclusion (that the benchmark is not saturated) robust even under this sample size.
>
> We have clarified this in the revised manuscript and supplementary materials, and now report additional details on the experiment setup in A.13.
>
> >Q3: Secondly, human users with RGB-D ego view only (not 3rd part FPS type view) cannot navigate complex paths as smoothly (action space) as a pre-trained agent in a similar environment. Requesting repeat of experiments in ego view only.
>
> Thank you for raising this point. In our evaluation, human participants and the RL agent operate under exactly the same conditions. Both receive only the egocentric RGB-D input and the same discrete action space, without any third-person view, global map, or intermediate representations. Humans do not have access to any additional information beyond what the agent receives. We have clarified this in the revised manuscript and demonstrated a test setup in Figure 10. Supplementary materials A.13.
>
> Regarding the concern about navigating complex paths, the objective of the human baseline is not to match the agent’s motion smoothness, but to determine whether skilled users can outperform the learned policy under identical perceptual and action constraints. All participants were able to do so consistently, indicating that the benchmark is not saturated.
>
> To ensure full transparency, we have also released the human evaluation interface and implementation in an anonymous GitHub repository at https://anonymous.4open.science/r/What-matters-in-RL-objnav-humanist-9D01,  allowing reviewers and the community to directly verify the test setup.

---

> ### Author Response · Authors · 2025-11-24
> **Official Comment by Author (4/4)**
>
> >Q4: In table 3, there is sudden jump in success for applying heuristics for obstacle avoidance helper.
>
> Thank you for carefully examining the results. We are not entirely certain which specific part of Table 3 the reviewer is referring to. We do not use an “obstacle avoidance helper” in our system, so we assume the question is about the untrapping helper, whose goal is to help the agent recover from narrow or stuck situations (not specifically obstacle avoidance).
> We further assume that the “sudden jump” refers to the approximately 3% increase in success rate when this helper is activated (difference between the first and third row), we would like to clarify that this improvement is not abnormal. The baseline success rate is already relatively high, so even a few additional successful recoveries can result in a noticeable relative gain. This also aligns with our claim that test-time enhancements can have a meaningful positive impact on performance.
> We have double-checked the numbers to ensure there were no mistakes in computation or reporting, and the values are correct. If we have misunderstood the reviewer’s concern, we would appreciate more specific clarification so that we can address it more directly and continue the discussion productively.
>
> >Q5: Line 941: How can agent come out if it is stuck at a corner and is rotating in a loop when dynamic goals swap by oscillation?
>
> Our goal-switching logic is primarily designed to handle situations where the current goal becomes unreachable, which may occur for several reasons, not only because the agent is stuck, but also due to perception and mapping errors or because a region has already been fully explored and can no longer be expanded.
> When the agent is trapped in a narrow space, the untrapping helper will attempt to execute an escape maneuver. If the untrapping helper is not enabled, switching between goals can still help the agent break out of rotational loops by repeatedly replanning to different locations. This is why we also include an extended time limit before switching goals, allowing the planner enough time to attempt a new path that may lead the agent out of the corner.
>
>
> >Q6: Instead of quite old Sem-Exp type map, probably two extreme ends could be tested - like simple semantic 3D voxel map and 3DLLM map [https://arxiv.org/pdf/2307.12981] to detect the prowess of map representation in downstream tasks and overhead.
>
> Compared with 2D semantic maps, 3D voxel maps contain height information about objects and obstacles, which could be beneficial for navigation. However, the change in observation dimensionality from 2D to 3D will greatly increase the processing time of each environment interaction. RL approaches typically require millions of interactions to achieve good performance, making this increase particularly costly.
>
> To verify whether height information actually benefits navigation, we conducted an experiment using a map that stores height information. The results can be found in Appendix 7.

---

> ### Author Response · Authors · 2025-11-27
> **A Gentle Reminder**
>
> Dear reviewer xY4S, thank you once again for your time and thoughtful review. We sincerely appreciate your feedback!
>
> The end of the discussion period is rapidly approaching, and we would greatly appreciate it if you could review our response and let us know whether your concerns are adequately addressed. If not, or in case you have any further concerns, we would be more than happy to work with you on improving the paper.

---

### Official Review · Reviewer_eFcx · 2025-11-15

**Soundness:** 4
**Presentation:** 3
**Contribution:** 3
**Rating:** 4
**Confidence:** 2

**Summary:**

The paper examines Object-Goal Navigation (ObjectNav), in which a robot must find target objects in previously unseen environments using only onboard perception. Noting the lack of a unified analysis of what drives performance, the authors conduct a large-scale empirical study of modular RL-based ObjectNav systems, decomposing them into three components: perception, policy, and test-time enhancement. Their findings show that perception quality and test-time strategies are the primary performance drivers, while policy improvements with current methods yield only marginal gains.

**Strengths:**

Investigating the contributions of different components in RL is very interesting.

The paper offers practical recommendations for improving ObjectNav performance.

The inclusion of a human baseline is also valuable, clearly illustrating the performance gap between current RL-based systems and human-level navigation.

**Weaknesses:**

The paper uses Habitat as the testbed, but it appears that only a single dataset is employed for experiments. It would strengthen the paper to include evaluations on additional datasets to validate the generality of the findings.

More details should be provided about the three essential modules—perception, policy, and test-time enhancement—including how each is implemented, trained, and evaluated. The descriptions are currently high-level and could benefit from clearer examples or pseudo-code.

In the Introduction, Figure 1 is referenced conceptually but not explicitly mentioned in the text. It may be better to move Figure 1 into the Introduction or clearly cite it there, as it provides an overview of the framework that helps readers understand the study structure early on.

The Test-time Enhancement Module Design Choices section needs clarification. It is not entirely clear what “test-time enhancement” means—whether it refers to inference-time heuristics, post-processing strategies, or adaptive decision rules applied without retraining. Additionally, Figure 2 does not explicitly show the four failure mode categories discussed in the text: (a) trapping in narrow spaces, (b) object misidentification, (c) repeated exploration, and (d) map overlap across floors due to undetected staircases. The figure should be updated or annotated to illustrate these cases for clarity.

The paper analyses the modules within the RL framework but uses heuristic rules for navigation instead of a learned neural policy conditioned on the top-down semantic map. The motivation for this design choice should be explained

**Questions:**

What are the details of the Human Experts—for example, who are they, what is their background, and how do they operate during the evaluation?

---

> ### Author Response · Authors · 2025-11-21
> **Official Comment by Authors (1/3)**
>
> Thank you for the insightful review and questions. We have submitted a revised version addressing your comments. Please find our detailed responses below:
>
> >W1: The paper uses Habitat as the testbed, but it appears that only a single dataset is employed for experiments. It would strengthen the paper to include evaluations on additional datasets to validate the generality of the findings.
>
> Thank you for the helpful suggestion regarding dataset diversity. We used Habitat with the Gibson dataset for all experiments. We agree that evaluating additional datasets would strengthen the generality of our findings.
>
> Following your advice, we have added a new set of experiments on the Habitat Matterport 3D (HM3D) dataset [1] , which is also supported by Habitat. We randomly chose five scenes from the test set of HM3D. And we conducted multiple trials across different perception settings and test-time enhancement combinations. The results on HM3D exhibit similar performance trends as those observed on Gibson, further validating our conclusions about the contribution of each module and demonstrating the generality of our findings across environments. We appreciate the insightful feedback. A summary of the key results is included below, and the full set of experiments is provided in Table 1 and Table 5.
>
>
> | Det.          | FPS | SR (%) | SPL (%) | D-SR (%) | D-SPL (%) |
> |---------------|-----|--------|---------|----------|-----------|
> | MRCNN         | 21  | 47.8   | 21.3    | 37.6     | 20.2      |
> | YOLO11-N      | 25  | 47.4   | 18.9    | 31.4     | 17.2      |
> | YOLO11-XL     | 23  | 48.0   | 19.0    | 32.4     | 17.0      |
> | RF-DETR-Seg   | 22  | 52.4   | 24.2    | 39.6     | 22.4      |
> | RF-DETR+SAM2  | 15  | 50.6   | 22.4    | 38.4     | 20.8      |
>
> Overall performance on HM3D is lower than on Gibson, which also aligns with results from other works [2]. Nevertheless, the main conclusions remain unchanged: more advanced perception models consistently lead to improved performance. For instance, RF-DETR outperforms Mask R-CNN. We also noticed that the RF-DETR segmentation variant performs better than RF-DETR+SAM2, which tends to over-segment without dedicated tuning.
>
> Finally, we also explored using large open-vocabulary models such as GroundingDINO. However, obtaining bounding boxes requires multiple inference rounds per query, reducing the system’s FPS to ~1 Hz, which is infeasible for both inference and training in our setting. We appreciate the reviewer’s insightful feedback and believe the new HM3D results significantly strengthen the submission. In addition, we now include HM3D in our benchmark suite for both training and evaluation to improve our framework’s dataset diversity. The full HM3D setup is also available in the anonymous GitHub repository, allowing the community to directly test additional modules or algorithms on diverse datasets.
>
> >W2: More details should be provided about the three essential modules—perception, policy, and test-time enhancement—including how each is implemented, trained, and evaluated. The descriptions are currently high-level and could benefit from clearer examples or pseudo-code.
>
> Thank you for the comment. We agree that providing additional implementation detail enhances clarity. We have been planning on releasing the full framework publicly. And we also prepared an anonymous GitHub repository to ensure transparent access during review. It contains full implementations of the perception, policy, and test-time enhancement modules, along with configuration files, training scripts, evaluation code, and illustrative examples:
>
> https://anonymous.4open.science/r/What-matters-in-RL-ObjNav-84A3
>
> We also provide an anonymous repository for the human evaluation interface to fully document the setup used in our human baseline study:
>
> https://anonymous.4open.science/r/What-matters-in-RL-objnav-humanist-9D01
>
> These repositories complement the paper by supplying detailed implementation resources, while the paper itself focuses on the higher-level design principles and key takeaways. Together, they ensure that the full pipeline is both transparent and easy to reproduce. We are continuing to work on refining the repository to make it more flexible to change configurations, with the ultimate goal of making it an open-source framework for the community.
>
> >W3: In the Introduction, Figure 1 is referenced conceptually but not explicitly mentioned in the text. It may be better to move Figure 1 into the Introduction or clearly cite it there, as it provides an overview of the framework that helps readers understand the study structure early on.
>
> Thank you for the suggestion. We have added an explicit reference to Figure 1 in the revised manuscript.
>
> [1] Habitat-Matterport 3D Dataset (HM3D): 1000 Large-scale 3D Environments for Embodied AI
>
> [2] Naviformer: A spatio-temporal context-aware transformer for object navigation

---

> ### Author Response · Authors · 2025-11-21
> **Official Comment by Authors (2/3)**
>
> >W4: The Test-time Enhancement Module Design Choices section needs clarification. It is not entirely clear what “test-time enhancement” means—whether it refers to inference-time heuristics, post-processing strategies, or adaptive decision rules applied without retraining.
>
> Test-time enhancement in our framework refers to inference-time post-processing strategies that operate without retraining the policy. These strategies act on top of the learned policy during execution and adjust behavior through lightweight rules or corrections. They are neither part of the training procedure nor learned components. Instead, they modify goal selection, map handling, or action decisions at run time to address failure patterns that commonly appear during deployment. Many of such enhancements are not reported in the original paper but are used very frequently in the code implementation. Hence, we found it useful to discuss these modules in detail. In the paper, we introduced three such strategies in the Test-time Enhancement Module section and describe them in detail in Appendix A.6.
>
> We thank the reviewer for pointing this out. We have added clearer explanations in the revised manuscript, specifically in Section 3 (System Overview), to address this point.
>
> >W5: Additionally, Figure 2 does not explicitly show the four failure mode categories discussed in the text: (a) trapping in narrow spaces, (b) object misidentification, (c) repeated exploration, and (d) map overlap across floors due to undetected staircases. The figure should be updated or annotated to illustrate these cases for clarity.
>
> Thank you for the suggestion. We have updated Figure 2 in the revised manuscript to explicitly include these major failure modes.
>
> >W6: The paper analyses the modules within the RL framework but uses heuristic rules for navigation instead of a learned neural policy conditioned on the top-down semantic map. The motivation for this design choice should be explained
>
> Thank you for the insightful comment. Our design choice is motivated by two considerations. First, our focus is on analyzing the contribution of each modular learning component within the RL framework. Prior study (e.g., [1]) has shown that modular learning tends to transfer better to real-world settings than end-to-end policies, and goal selection is a standard design in modular RL navigation systems (e.g., [2–5]).  Second, although it is possible to combine our goal-selection module with a learned low-level policy such as DD-PPO [6], we intentionally use non-learning methods, such as Habitat’s built-in non-learning local planner. This choice reduces confounding factors from learned velocity control policies and ensures that differences in performance can be attributed directly to the modules we study rather than to the variability of low-level navigation learning. This also provides a fair and consistent comparison across methods, as many related works employ the same Habitat local planner.
>
> Importantly, this design is not a limitation of our framework: the proposed module designs are compatible with learned low-level navigation policies, and in practice, one could readily switch to such approaches. Our goal in the paper is to isolate and evaluate the contribution of the modular components themselves.
>
> We hope this clarifies the reasoning behind our design choice, and we would be happy to further adjust the explanation in the revision.
>
> [1] Navigating to objects in the real world
>
> [2] 3d- aware object goal navigation via simultaneous exploration and identification
>
> [3] Stubborn: A strong baseline for indoor object navigation
>
> [4] Naviformer: A spatio-temporal context-aware transformer for object navigation
>
> [5] Frontier semantic exploration for visual target navigation
>
> [6] Dd-ppo: Learning near-perfect pointgoal navigators from 2.5 billion frames

---

> ### Author Response · Authors · 2025-11-21
> **Official Comment by Authors (3/3)**
>
> >Q: What are the details of the Human Experts—for example, who are they, what is their background, and how do they operate during the evaluation?
>
> Thank you for the comment. As noted in the original paper in Section A.10 of the Appendix, the human experts are robotics researchers with no prior experience in ObjectNav, intentionally selected to evaluate whether current benchmarks are saturated rather than to optimize human performance.
>
> At the same time, we do agree that the experimental setup of the human benchmark should be documented more clearly. Hence, we have now included additional details in the revised supplementary document A.10, and we can also include a screenshot of the user study interface to further clarify the evaluation process.
>
> We hope this clarifies the questions on the human expert evaluation process.

---

> ### Author Response · Authors · 2025-11-27
> **A Gentle Reminder**
>
> Dear reviewer eFcx, thank you once again for your time and thoughtful review. We sincerely appreciate your feedback!
>
> The end of the discussion period is rapidly approaching, and we would greatly appreciate it if you could review our response and let us know whether your concerns are adequately addressed. If not, or in case you have any further concerns, we would be more than happy to work with you on improving the paper.

---

### Author Response · Authors · 2025-11-24
**General Response from Authors**

Dear AC and reviewers,

We sincerely thank the reviewers for their valuable feedback regarding clarity, missing related work, and requests for additional ablation and baseline experiments. We also appreciate the reviewers who explicitly recognized the usefulness and relevance of our study for the ObjectNav and embodied AI communities.
In response to all comments, we have substantially improved the manuscript (with revisions highlighted in blue). The key updates are summarized below:

- Ablation Studies:
  - We include additional experiments using more recent object detectors.
  - We analyze the impact of incorporating the geometry information.
  - We evaluate different frontier-selection configurations.

- Related Work:
  - We have added all the references suggested by the reviewers and expanded our related-work discussion to better contextualize and distinguish our contributions.

- Test Dataset:
  - We include new experiments conducted on the HM3D dataset.

- Technical Clarity:
  -  We improved the clarity of both the methodology and experimental sections by restructuring the content, refining descriptions, and adding further technical details.

- Qualitative Results:
  - We provide additional qualitative results to better illustrate the effectiveness of the three test-time enhancement modules.

- Anonymous GitHub of Proposed Framework
  - To support clearer understanding, reproducible evaluation, and further improvement of the framework, we release our full training and evaluation codebase at: https://anonymous.4open.science/r/What-matters-in-RL-ObjNav-84A3 and the human baseline code at: https://anonymous.4open.science/r/What-matters-in-RL-objnav-humanist-9D01

We look forward to further engaging in discussions and appreciate the opportunity to refine our work based on the reviewers' constructive feedback.

---

### Meta-Review · Area_Chair_wSpL · 2025-12-23

**Summary:**

The paper received divergent ratings from the reviewers (6,6,4,2). While the reviewers appreciated the study and the comprehensive experimental analysis, they raised various concerns such as:

(1) Lack of clear descriptions for the three components,\
(2) Using only one benchmark for experiments,\
(3) Modular architectures being covered by prior work,\
(4) Too strong claims for the limited setting,\
(5) No exploration of different frontier selection methods,\
(6) Relying on old object detectors for studying the perception module.

The rebuttal addressed some of the issues. For instance, to address (3), it explained that the contribution of this work is studying modular methods rather than proposing new methods. Also, to address (6), new experiments with new object detectors are provided.

The AC carefully read the paper, the reviews and author responses. Some major issues still stand out. Most of the conclusions are already known. For instance, “improving the object detector yields significant gains in overall navigation performance” is a known fact. As another example, “Map augmentation generally leads to better performance compared to not using it.” is true by definition. Otherwise, it would not be introduced. Another example is “adding a reward for moving closer to the target object encourages the agent to make more efficient decisions, not only for exploration but also for approaching the target.” It is also known that a dense reward function is better than a sparse reward function.

Another issue is the strong claims of the paper. For instance, the abstract says “policy improvements with current methods yield only marginal gains”, but the conclusion is drawn from a fairly limited set of experiments. There are various new developments in the field compared to what has been used in the cited works.

The weakest part of the paper is the “test-time enhancement module.” As reviewer xY4S mentioned, they are a set of hacks to recover from the failures of perception or policy modules. The policy was not able to learn these to recover from failures. So, there is room to improve the policy contrary to what was mentioned in previous conclusions.

Given these issues, rejection is recommended. Additionally, prior work on the analysis of the ObjectNav task (e.g., ‘What Do Navigation Agents Learn About Their Environment?’, CVPR 2022) could provide valuable inspiration for strengthening future iterations of this paper.

**Reviewer Concerns:**

Please refer to the explanation provided above.

**Reviewer Scores:**

Reviewer eFcx: The reviewer needed further details about the modules. The required improvements are beyond the scope of the rebuttal and require a major rewrite. The concern about the rule-based heuristics is not addressed well either. So, no score change would be expected.

Reviewer xY4S: The concerns are addressed reasonably. An increase in the score would be possible if the reviewer considers only their own review.

Reviewer ehQz: The concerns are about the known conclusions and strong claims without evidence still persist. The rebuttal does not address them well. Potentially, a lower score would be expected.

Reviewer HZyA: The reviewer requested additional elaboration on several aspects, which the rebuttal has provided. However, the response does not introduce substantive changes that would justify a score change.

---

### Decision · Program_Chairs · 2026-01-26

Reject